


# Implications of Model Selection: A Comparison of Publicly Available, CONUS-Extent Hydrologic Component Estimates

Samuel Saxe[1,2], William Farmer[1], Jessica Driscoll[1], Terri S. Hogue[2,3]

[1]Analysis and Prediction Branch, U.S. Geological Survey, Lakewood, CO 80225, U.S.
[2]Hydrologic Science and Engineering, Colorado School of Mines, Golden, CO 80401, U.S.
[3]Department of Civil and Environmental Engineering, Colorado School of Mines, Golden, CO 80401, U.S.

*Correspondence to*: Samuel Saxe (ssaxe@usgs.gov)

**Abstract.** Spatiotemporally continuous estimates of the hydrologic cycle are often generated through hydrologic modeling, reanalysis, or remote sensing methods, and commonly applied as a supplement to, or a substitute for, in-situ measurements
when observational data are sparse or unavailable. Many of these datasets are shared within the public domain, helping to accelerate progress in the fields of hydrology, climatology, and meteorology by (a) reducing the need for technical programming skills and computational power, and (b) providing a wide range of forecast and hindcast estimates of terrestrial hydrology that can be applied within ensemble analyses. Past model inter-comparisons focused on the causes of model disagreement, emphasizing forcing data, model structure, and calibration methods. Despite the relatively recent increased
application of publicly available modeled estimates in the scientific community, there is limited discussion or understanding of how selection of one dataset over others can affect study results. This study compares estimates of precipitation (P), actual evapotranspiration (AET), runoff (R), snow water equivalent (SWE), and rootzone soil moisture (RZSM) from 87 unique datasets generated by 47 hydrologic models, reanalysis datasets, and remote sensing products at the monthly timescale across the conterminous United States (CONUS) from 1982 to 2014. To understand the effect of model selection on terrestrial
hydrology analyses, 2,925 water budgets were calculated over 2001-2010 for each of eight Environmental Protection Agency ecoregions by iterating through all combinations of 43 hydrologic flux estimates.

Variability between hydrologic component estimates was shown to be higher in the western CONUS, with median coefficient of variation (CV) ranging from 11-22% for P, 14-27% for AET, 28-153% for R, 92-102% for SWE, and 39-92% for RZSM. Variability between estimates was lower in the eastern CONUS, with median CV ranging from 5-15% for P, 13-
23% for AET, 29-96% for R, 64-70% for SWE, and 44-81% for RZSM. Inter-annual trends in estimates from 1982-2010 show more comprehensive agreement for trends in P and AET fluxes but common disagreement for trends in R, SWE, and RZSM. Correlating fluxes and stores against remote sensing-derived products shows poor overall correlation in the western CONUS for AET and RZSM estimates. Iterative budget relative imbalances were shown to range from -50% to +50% in major eastern ecoregions and -150% to +60% in western ecoregions, depending on models selected. These results
demonstrate that disagreement between estimates can be substantial, sometimes exceeding the magnitude of the measurements themselves. The authors conclude that multi-model ensembles are not only useful, but are in fact a necessity,





to accurately represent uncertainty in research results. Spatial biases of model disagreement values in the western United States show that targeted research efforts in arid and semi-arid water-limited regions are warranted, with the greatest emphasis on storage and runoff components, to better describe complexities of the terrestrial hydrologic system and
reconcile model disagreement.

## 1 Introduction

A long-term goal of the atmospheric and hydrologic scientific communities has been to produce accurate estimates of the hydrologic cycle across continental and global domains (Archfield et al., 2015; Beven, 2006; Freeze and Harlan, 1969). Various methodologies have been applied to meet this goal, typically in the form of physically based, reanalysis, or remote
sensing-based models. Many of the resulting estimates are made publicly available by an assortment of scientific entities at both continental- and global-extents, across a wide range of spatiotemporal resolutions. These datasets accelerate progress in the atmospheric and hydrologic sciences by filling knowledge gaps in data-sparse regions, reducing the need for time-consuming and computationally expensive modeling, and by providing numerous estimates to apply within ensemble analyses.

Publicly available modeled estimates have been applied to work on water budget analyses (Gao et al., 2010; Pan et al., 2012; Rodell et al., 2015; Smith and Kummerow, 2013; Velpuri et al., 2019; Zhang et al., 2018), effects of climate change (LaFontaine et al., 2015; G. J. McCabe et al., 2017), and water availability and use (Landerer and Swenson, 2012; Thomas and Famiglietti, 2019; Voss et al., 2013; Zaussinger et al., 2019). Rapid increases in computational power and data accessibility following the advent of early global- and continental-extent hydrologic models in the 1980s and 1990s (Koster
and Suarez, 1992; Manabe, 1969; Sellers et al., 1986; Yang and Dickinson, 1996), and increasingly higher-resolution passive and active satellite measurements of both the surface and subsurface of the earth (Alsdorf et al., 2007; M. F. McCabe et al., 2017), have led to an explosion in the production of multi-decadal, continental- to global-extent models estimating all major components of the hydrologic cycle (Peters-Lidard et al., 2018). Accordingly, this has given rise to a multitude of model comparison and evaluation projects throughout the scientific literature.

Model evaluation studies can be loosely categorized as one of: (a) Model Foundations & Justifications (MFJs); b) Model Intercomparison Projects (MIPs); or c) Current State Analyses (CSAs). The reader is referred to Table 1 (Methods) and Appendix 1 for model abbreviation definitions, descriptions, and references discussed in this paper. MFJs are presentations of novel products designed to increase the accuracy and understanding of one or more components of the hydrologic cycle, created by the originators of the novel product. These typically present the structural equations and required parameters of a
model along with a quantitative evaluation of how, or if, the model demonstrates significant improvement relative to earlier products, often divided into several publications. MIPs are defined here as formal collaboration projects aimed to attribute differences in model process representation to differences in model design by operating multiple models at identical or



similar spatiotemporal scales using constant meteorological forcings, geophysical parameterizations, and calibration

schemes. Collaborators in MIPs often include numerous scientists with vested interests in the analyzed models, contributing

expertise in model structure and development. Some examples of these include the early Coupled Model Intercomparison

Project (CMIP) (Covey et al., 2003), the Integrated Hydrologic Model Intercomparison Project (IH-MIP) (Kollet et al., 2017;

Maxwell et al., 2014), the Agricultural Model Intercomparison and Improvement Project (AgMIP) (Rosenzweig et al.,

2013), and the Water Model Intercomparison Project (WaterMIP) (Haddeland et al., 2011).

CSAs, into which this paper falls, analyze externally generated model estimates across a range of spatiotemporal scales to

provide a comprehensive overview of the current knowledge within the scientific community of hydrologic processes.

Authors of CSAs are usually not involved in development of the subject models and thus provide an independent and

unbiased analyses. Past evaluation studies have compared model process representations of flux (precipitation,

evapotranspiration, runoff) and storage (snow water equivalent, soil moisture) terms across an assortment of spatiotemporal

resolutions and climatic zones, often validating model estimates against in-situ measurements to identify the best performing

model.

The early Third Precipitation Intercomparison Project (PIP-3) from Adler et al. (2001) compared and validated 31 satellite-

based, reanalysis, and climatological experimental and operational precipitation (P) products for 1 year across both ocean

and land surfaces. PIP-3 found high disagreement between models, noting greatest performance from merged remote

sensing-rain gauge products. Daly et al. (2008) presented the widely used operational Parameter-elevation Relationships on

Independent Slopes Model (PRISM) and compared validation metrics against since-deprecated versions of Daymet

(Thornton et al., 1997) and WorldClim (Hijmans et al., 2005), showing substantial improvement, especially in

topographically complex and data sparse regions. Guirguis and Avissar (2008) found that the majority of variability between

nine P products in the western conterminous United States (CONUS) was caused by the incorporation of P values from

mountainous regions, cold-season gradients, monsoon signals, and arid region spring and summer precipitation. Similarly,

Derin and Yilmaz (2014) evaluated four satellite-derived P estimates against an interpolated gauge-based product and

showed that the satellite products did not accurately represent P gradients along mountainous terrain due to underestimation

(overestimation) of windward (leeward) P.

Donat et al. (2014) validated extreme daily P estimates from gauge-based and reanalysis datasets and showed high

disagreement in magnitude but good correlation between reanalyses and observations in North America. Agreement between

models from the same modeling centers was higher than with other products, indicating similarities in modeling techniques.

Prat and Nelson (2015) validated four satellite, radar, and gauge-based P estimates across the CONUS, noting generally high

agreement between models at the annual scale but poor performance in the high P regions in the western CONUS when

validated against rain gauge measurements. The most current and comprehensive CSA, from Sun et al. (2018), compared 30

gauge-based, reanalysis, and remote sensing (RS) global P datasets at varying spatiotemporal scales. Their results indicated





that variability was as high as 300 mm/year and reanalysis products showed the greatest overall variability by category. Spatially, variability was greater in regions of either higher annual P or complex topography. Specific to the CONUS, inter-product variability appeared to be greater in the basin-and-range and Pacific Northwest regions than in eastern regions.

Evaluation and comparison literature focused on actual evapotranspiration (AET) estimates is plentiful, albeit often limited in scope. The CSA by Mueller et al. (2011) compared numerous global AET estimates from observational and land surface

model (LSM) sources and showed global standard deviations between products ranging from 58 to 73 mm/year, finding that LSMs produce lower mean global AET than observation-derived and reanalysis-incorporated datasets. However, this previously comprehensive study is now arguably outdated as it does not sufficiently describe the modern state of knowledge within the AET modeling community. Many of the LSM outputs used in the Mueller et al. (2011) study are no longer publicly available or are not within popular use among the scientific community, and none of the modern RS models are

analyzed. The MFJs from Senay et al. (2011) and Velpuri et al. (2013) validated the operational Simplified Surface Energy Balance (SSEBop) AET RS model (Senay et al., 2013, 2007) against in-situ eddy covariance AET and the Mapping EvapoTranspiration at high Resolution with Internalized Calibration (METRIC) RS model (Allen et al., 2007), respectively, showing that SSEBop performed favorably against in-situ observations and yielded improved estimates in less complex topography than the more difficult and expensive METRIC. Carter et al. (2018) evaluated AET from the Noah LSM (Y. Xia

et al., 2012a), the MOD16-A2 RS model (Mu et al., 2007), and the ALEXI RS model (Anderson et al., 1997) against water balance AET (WBET) at more than 600 gauged basins across the CONUS and found that the RS estimates overpredicted in western and Appalachian basins while the Noah LSM underpredicted in eastern regions. All models showed highest disagreement in Pacific and eastern basins and underestimated WBET by more than 50% in arid regions of the central CONUS. Differences in WBET and RS AET were significantly correlated with basin slope which the authors concluded was

caused by negative biases of precipitation estimates in high-slope regions. McCabe et al. (2016), as part of the GEWEX LandFlux project, compared the Global Land Evaporation Model (GLEAM), Penman-Monteith equation (PM-Mu), Priestly-Taylor Jet Propulsion Laboratory (PT-JPL), and Surface Energy Balance System (SEBS) methods for estimating AET from flux tower observations and found that PT-JPL and GLEAM performed the best overall, and PM-Mu (SEBS) underestimates (overestimates), though compared by biome no single method significantly outperformed any of the others.

While evaluation and comparison studies of large-domain modeled P and AET are relatively common, literature on runoff (R) is much sparser. Modeling of the complete hydrologic cycle at continental- and global-extents is most commonly performed using LSMs, often with the goal of generating lower boundary conditions for atmospheric or meteorological models and thereby targeting the vertical AET flux with less emphasis or interest in 2-dimensional runoff processes (Archfield et al., 2015). However, interest in LSM R estimates has grown over the last decade, resulting in several multi-

model evaluation studies. The WaterMIP (Haddeland et al., 2011) compared six LSMs and five global hydrologic models (GHMs) and showed that LSMs underestimated R partitioning and snow water equivalence (SWE) relative to GHMs,



attributed to differences in snow process representation. The authors agreed with previous findings that conclusions should not be drawn from a single model realization due to the uncertainty associated with model differences. Similarly, Xia et al. (2012b, 2012a), presenting the second phase of the North American Land Data Assimilation System (NLDAS2), compared

and validated R estimates from four LSMs across the CONUS. The authors attributed general model differences in the northeast and western mountainous regions of the CONUS to cold season processes. Differences in subsurface R and evaporation were attributed to varying sub-surface hydrology process representation. Validation of streamflow and AET against observational streamgage and WBET data at hundreds of small basins and eight major basins showed that the Mosaic LSM underestimated mean annual R in much of the CONUS while the Noah LSM simulated R well in much of the eastern

CONUS but overestimated in the southeast and Midwest. The authors noted that an ensemble mean of the four models demonstrated the best overall performance.

Various studies have identified common shortcomings of SWE modeling, with estimates from RS, hydrologic, reanalysis, and snow-specific models almost universally underestimating SWE depth across much of the CONUS and performing especially poorly in forested regions (Broxton et al., 2016; Chen et al., 2014; Dawson et al., 2018, 2016; Essery et al., 2009;

Mudryk et al., 2015; Rutter et al., 2009; Vuyovich et al., 2014).   Discrepancies arise in analyses of SWE accumulation/ablation timing and in attributing causes of inter-model disagreement. Broxton et al. (2016) noted that LSMs produce earlier snowmelt timing than an observation-based reference product, while the CSA from Vuyovitch et al. (2014) found that RS model SWE timing tended to agree with SNODAS values. Phase two of the Snow Model Intercomparison Project (SnowMIP2) (Essery et al., 2009; Rutter et al., 2009) found that models, including LSMs and snow-specific models,

tended to yield later ablation timing.   Broxton et al. (2016) also suggested that LSM SWE uncertainty may be due to differences in model structure, while the CSA from Mudryk et al. (2015) argued more specifically that climatology variability and inter-model calibration errors of LSMs are more strongly controlled by model structure and meteorological forcing, respectively.   No single LSM has been shown to consistently outperform any single other LSM, and the ensemble mean of various model types actually underperformed relative to individual models (Chen et al., 2014; Rutter et al., 2009).

Comparison and evaluation of soil moisture (SM) models is often difficult due to inconsistencies in soil depths and reporting units between model output and observational measurements. Despite this, several studies compared LSM and RS modeled SM, evaluating against in-situ data when available.   Koster et al. (2009), as part of a larger discussion on applications of LSM and reanalysis SM, compared the detrended degree of saturation (root zone SM divided by total water holding capacity) between seven LSMs from the follow-on Global Soil Wetness Project (GSWP-2) (Dirmeyer et al., 2006), finding

substantial differences in both mean climatology and dynamic range. The authors argue, agreeing with previous work (Saleem and Salvucci, 2002), that LSM SM should not be considered a direct measure of actual soil water content but rather a "model-specific index of wetness". Instead, they advise that temporal correlation of SM time series should be the most accurate statistical method to evaluate modeled estimates. Applying this, the authors note higher r2 (approximately 0.60-



0.80), derived as the mean squared inter-correlation between models, in the southeastern, central, and southwestern CONUS

with poorer r2 (approximately 0.30-0.55) in northern regions (Koster et al., 2009, Fig. 5). Bands of lower agreement in the CONUS are shown in arid, desert regions of the CONUS. Brocca et al. (2011), also utilizing time-series correlation, validated RS SM at 17 sites across Europe and found generally good correlation that decreases with increasing vegetation density. However, presenting correlation values as validation statistics may not appropriately describe model skill (Legates and McCabe, 1999) and as such the above-mentioned research may not fully describe the scale of model disagreement. Xia

et al. (2014) validated four NLDAS-driven LSMs against in-situ observations across the CONUS and found that the LSMs were able to capture seasonal and annual variations but overestimated temporal variability. Further, all models were shown to underestimate absolute soil moisture and yielded large root mean squared error (RMSE) values, but an ensemble mean of the models outperformed any individual model.  They attribute low correlation metrics in arid regions to model difficulties in capturing lower magnitude changes, as opposed to high-precipitation eastern regions where effects on soil moisture are

larger.

Looking more generally by model type, Schellekens et al. (2017) identified lower model agreement between LSMs and GHMs in snow-dominated, tropical rainforest, and monsoon regions for primary flux components. Gao et al. (2010) and Sheffield et al. (2009) compared RS models as part of larger water budget studies and found that AET and terrestrial water storage (TWS) models showed greater agreement than P datasets, which the authors attributed to similarities in

methodologies. Gao et al. (2010), additionally including LSMs in comparison analyses, showed that LSMs overestimated AET relative to RS models, and that RS model error magnitudes are greatest among precipitation datasets.

Many of the papers discussed above have focused on validation against in-situ and ex-situ measurements. However, validation often yields unsatisfactory representations of model skill. Validation datasets are often spatiotemporally discontinuous and model estimates in grid structure may not sufficiently represent the sub-grid heterogeneity that is present

in point data, especially in topographically and ecologically complex regions. Further, observational measurements can have significant associated uncertainty (Baldassarre and Montanari, 2009).  Validation literature is heavily focused on comparison of error statistics and spatial summaries of model skill, rarely discussing the effect of model disagreement on research results.

This research seeks to supplement past comparison and validation literature by evaluating how utilization of publicly

available water budget component estimates can affect relevant studies where products are used. We differentiate this work from previous MIPs, MFJs, and CSAs by placing results in the context of multiple water budget analyses. We seek to quantify how estimates of component magnitudes and long-term trends differ in the CONUS, as well as within ecologically distinct regions.  The primary goal of this work is to quantify model disagreement in terms of magnitude, inter-annual trend, and correlation against remote sensing products.  The effects of model disagreement are shown in regional water budget

analyses.  We undertake a robust comparison of P, AET, R, SWE, and SM estimates generated through hydrologic models,



reanalysis datasets, and remote sensing products. We iteratively calculate water budget imbalances in eight regions, by applying a range of flux estimates to quantify how model selection may impact residuals.

## 3 Methods and Data

### 3.1 Data Categories

Hydrologic estimates are divided into water budget flux components of P, AET, and R, and storage components of SWE and RZSM), representing the primary fluxes and stores of a surface water budget. Datasets were selected by prioritizing public availability, ease of access, and relative use within the research community. These datasets are subdivided into loosely defined categories of hydrologic models, reanalysis datasets, and RS-derived products. References and spatiotemporal information for each dataset are provided in Table 1 and more detailed long-form descriptions are provided in Appendix 1.

### 3.1.1 Hydrologic Models

The hydrologic model category (Table 1, Appendix 1) includes any estimates generated using equations or concepts attempting to represent real-world hydrology. Model output differences are strongly controlled by forcing datasets (Elsner et al., 2014; Mizukami et al., 2014), calibration methods (Mendoza et al., 2015), applied equations (Clark et al., 2015a), model structure (Clark et al., 2015a, 2015b), and geophysical parameter availability (Beven, 2002; Bierkens, 2015; Mizukami et al.,
2017). The most common hydrologic models this study evaluates are conceptual and physically based. Conceptual models derive terrestrial hydrology estimates through empirical relationships between fluxes and stores, typically through a water balance model (WBM) in the style of Thornthwaite (1948) (NHM-MWBM and TerraClimate). Physically based models utilize meteorological forcing data and solve equations describing physical conservation laws of mass, energy, and momentum. These can be further sub-divided by targeted hydrologic variables: land surface models (LSMs) target land-
atmosphere interactions, especially AET, while catchment models (CMs) target streamflow (Archfield et al., 2015). Additionally, LSMs are often one-dimensional models that operate at discrete spatial intervals, typically grid cells, lacking horizontal transfer of surface or subsurface water between regions. CMs, on the other hand, utilize two-dimensional model structures by routing overland and subsurface flows between spatial domains to better realize surface and groundwater estimates using prescribed stream networks. The CLSM, described in the literature as a land surface model (Koster et al.,
2000), is discussed in conjunction with the NHM-PRMS because of the sub-grid catchment network used to model horizontal runoff and streamflow fluxes. LSMs are the most common hydrologic model here (e.g. CLM, H-TESSEL, Mosaic, Noah, SiB, and VIC) because they are often coupled with global-extent atmospheric research and thus more commonly operated at the spatial extent of this study, while CMs are more typically operated at the catchment or basin scale. Additional products included in the hydrologic model category are those using simplified WBMs falling outside the
conceptual model paradigm (CPC, CSIRO-PML, GLEAM, and VegET) and those using component-specific physically based models (SNODAS).





### 3.1.2 Reanalysis

Reanalysis datasets (Table 1, Appendix 1) assimilate multi-source in-situ and ex-situ observational data into spatiotemporally continuous four-dimensional estimates of continental- or global-scale atmospheric and meteorological fluxes using numerical algorithms. This category includes reanalysis datasets derived solely from in-situ and ex-situ data (e.g. CMAP, DayMET, Maurer et al. 2002, UoD-v5), as well as those assimilating or blending multiple reanalysis products with or without observational measurements (e.g. CanSISE, gridMET, Livneh et al. 2013, NLDAS2). In past studies, reanalysis models were often grouped and defined separately from gridded datasets derived through statistical interpolation of in-situ observations (e.g. WaterWatch, Reitz et al. 2017, GPCC). We combine the two categories, defining the single reanalysis category as containing products not generated solely from remote sensing observations or through terrestrial hydrologic models. The reasons for this are two-fold: (1) to simplify reporting of results and discussion across water budget components, and (2) although interpolation-based products are often used as reference datasets in model validation studies, especially for precipitation, accuracy decreases in regions with sparse observations and complex topography (Hofstra et al., 2008) and so should be considered modeled products.

### 3.1.3 Remote Sensing-Derived

The RS-derived datasets (Table 1, Appendix 1) are components of the hydrologic system derived from passive and/or active ex-situ observations. We note the use of the term "RS-derived" rather than "RS" because none of the datasets used here can be truly described as direct observations, always relying on a range of modeling techniques to create hydrologic estimates. For example, the AMSR-E/Aqua SWE product uses a sequence of steps including a snow detection routine, followed by a physical retrieval algorithm, which is then validated against observational SWE data (Chang and Rango, 2000), and the SSEBop product uses a radiation-driven energy balance model to estimate AET (Senay et al., 2013). Remote sensing datasets include estimates modeled from single-sensor ex-situ observations (MOD16-A2, SSEBop), though most utilize an assembly of information from both passive and active sensors at various spatiotemporal resolutions (e.g. ESA-CCI, GPCP-v3).

### 3.2 Data Processing

Datasets included in this research cover a broad spectrum of spatial resolutions, extents, coordinate reference systems, and time scales, and therefore required a uniform spatiotemporal system to better facilitate comparison. To that end, gridded (e.g. NLDAS2-Mosaic) and polygonal (e.g. NHM-PRMS) datasets were aggregated by area-weighted mean to 10 Environmental Protection Agency Ecoregions, Level I (Omernik and Griffith, 2014) that encompass the Watershed Boundary Dataset Hydrologic Units (U.S. Geological Survey, 2016) (Fig. 1, Table 2) over the CONUS. Datasets were processed in their native coordinate system to avoid interpolation of raw values and the reference system of the ecoregions spatial dataset was transformed to match that of each target dataset. Flux and storage terms were aggregated to monthly time





steps. Flux values (P, AET, and R) were summed from hourly or daily when necessary and storage values (SWE, RZSM) were averaged.

Units of P, AET, R, and SWE are uniformly presented as equivalent water depth in millimeters (mm). RZSM is provided by datasets as either equivalent water depth (mm) or volumetric soil moisture content (m3/m3), and denoted as RZSME or RZSMV, respectively. The RZSME and RZSMV categories are merged during results and discussion when magnitude is irrelevant, such as with long-term trend and correlation against RS products. All data generated through these methods are available in an associated data release hosted on the USGS ScienceBase (Saxe et al., 2020).

**3.3 Statistics**

Variability is measured in terms of standard deviation (SD) and coefficient of variation (CV):

$$CV = \frac{\sigma}{\bar{x}} \; x \; 100\% \tag{1}$$

where σ is the standard deviation across all products of a given component in a given water year and $\bar{x}$ is the associated mean. The SD and CV statistics were calculated for each water year (WY; a water year is the 12-month period October 1

through September 30 designated by the calendar year in which it ends) between available estimates for all components. Annual values of flux terms (P, AET, and R) were derived by summing monthly values to WYs, and storage terms were derived by averaging monthly values by WY. Incomplete WYs (n months < 12) were discarded. Because of incompatibility between dataset temporal ranges, SD and CV values were divided into two 15-WY periods: the early period consisting of WYs 1985-1999, and the late period consisting of WYs 2000-2014.

Variability in seasonal timing of SWE estimates was compared for trends of both accumulation and ablation in terms of relative timing, defined here as the difference between the antecedent month of snow accumulation or ablation for each dataset from the mean antecedent month of the remaining datasets, calculated for each Julian year from 1985-2014. The antecedent accumulation (ablation) month is defined as the first month of the June-December (January-May) period, where change in SWE is greater than (less than) 1 mm per month (-1 mm per month). Some datasets contained anomalies in

January and February, showing a large negative change in SWE followed by positive rates of SWE accumulation for 1-2 more months, which introduced a negative bias into relative timing values. To eliminate these biases, ablation timing was only considered for months subsequent to the final date of positive SWE rate of change. Resulting annual relative timing values are presented for each SWE product grouped by (a) early and late periods, (b) accumulation and ablation, and (c) ecoregions Northern Forests and Northwestern Forested Mountains. To simplify comparison of SWE timing between trends

and ecoregions, annual directions of relative timing were summarized by percent direction, defined here as the percentage of years with a positive relative timing value:


$$p^+ = \frac{|\mathbb{R}^+|}{|\mathbb{R}|} \; x \; 100\% \, , \tag{2}$$

where $p^+$ is percent direction, $|\mathbb{R}^+|$ is the cardinality (number of elements in a set) of positive, real relative timing values, and $|R|$ is the cardinality of all real values. Within the figure presenting summary values (Appendix 2.4), percent direction

values less than 50% (i.e. those dominantly negative) are converted to a scale of 50-100% with:

$$pD = \begin{cases} p^+ & p^+ > 50 \\ 100\% - p^+ & p^+ < 50 \end{cases} , \tag{3}$$

where *pD* is the directionalized percent direction colored according to the dominant direction value. Positive *pD* uses a green gradient and negative *pD* uses a purple gradient.

Both the Mann-Kendall  trend test ($\tau$) (Kendall, 1938) and Sen's slope estimator (Sen, 1968) are used to identify and

measure monotonic trends in annual values over WYs 1982-2010.  Trend significance is evaluated using p-values (p) in a binary significance test, assuming an alpha ($\alpha$) of 0.05 and a null hypothesis of no monotonic trend.  Under the condition of p $< \alpha$, the null hypothesis is rejected and the alternative hypothesis, that of the presence of a monotonic trend, is assumed; if p $> \alpha$, the null hypothesis is not rejected. Disagreement in the presence of significant trend and trend direction is quantified using the unalikeability coefficient (u) which measures how often categorical variables differ on a $0 \leq u \leq 1$ scale, with 0 and

1 being complete agreement and disagreement, respectively (Kader and Perry, 2007).  This study compares trends between 11 to 16 different products ($11 \leq n \leq 16$) depending on water budget component and uses categorical values (c) of (a) significant negative trend, (b) significant positive trend, and (c) no significant trend, derived from the direction (sign) and significance ($\alpha = 0.05$) of $\tau$.  When n > c, maximum possible u decreases from 1 exponentially to an asymptote of 2/3 as the ratio of n:c increases. With a n:c ranging from 11:3 to 16:3, maximum possible u in this trend analysis is approximately 0.71.

Spearman's rho (Spearman, 1904) is applied to quantify correlation between RS components P, AET, SWE, and RZSM and the analogous components of hydrologic models and remote sensing datasets. Spearman's rho ($\rho$), a non-parametric rank correlation metric, is used to estimate correlation, producing values on a -1 to 1 scale where 1 is perfect positive correlation and -1 is perfect negative correlation:

$$\rho = 1 - \frac{6 \sum d_i^2}{n(n^2 - 1)} \, , \tag{4}$$

where d is the difference between ranks and n is sample size. A binary significance test is used to test correlation statistics, assuming $\alpha = 0.05$ and a null hypothesis that there is no relationship between datasets.  If p $< \alpha$, the null hypothesis is rejected and a significant correlation between the datasets is assumed; if p $> \alpha$, the null hypothesis is not rejected.





Correlation is computed and assessed along the monthly timestep, requiring a minimum 48 months of temporal overlap between the modeled estimates and remote sensing dataset.

Water budgets are calculated assuming a steady-state system and solved for imbalances by:

$$P = AET + R + \varepsilon , \hspace{8cm} (5)$$

$$\varepsilon = P - AET - R , \hspace{8cm} (6)$$

where ε is imbalances. Imbalances, ε, cannot be accurately defined as residuals. In reality, ε is the sum of excluded fluxes, in addition to model uncertainty (residuals). Excluded fluxes include both natural (e.g. groundwater recharge, changes to
long-term storage) and anthropogenic (e.g. groundwater extraction) hydrologic processes. Relative imbalances (Rε) are calculated to better compare water budget results between regions of varying hydrologic flux by weighting water budget ε against the total input P:

$$R\varepsilon = (\varepsilon/P) \times 100\% , \hspace{7cm} (7)$$

Water budget relative imbalances were calculated from summed P, AET, R, and ε over the 10-water year period of 2001-
2010 for 15 P, 15 AET, and 13 R estimates (noted in Table 1) with temporally continuous monthly data for eight ecoregions. Each ecoregion yielded 2,925 water budgets by iterating through all possible combinations of models, totaling 23,400 water budgets in primary regions of the CONUS. The ecoregions Tropical Wet Forests and Southern Semiarid Highlands were excluded due to their small percentage area of the total study domain (0.27% and 0.82%, respectively) (Table 2). In addition, a single water budget was calculated for each of the eight ecoregions using the ensemble means of the model estimates over
the same period.

## 4 Results

### 4.1 CONUS Domain Average

### 4.1.1 Magnitude Variability

Variability between annual estimates of states and fluxes of the hydrologic cycle are substantial when averaged over the
CONUS for each studied component (Fig. 2). The actual magnitude of model estimate differences averaged by water budget component, measured as SD, is similar for P, AET, and R (Fig. 2a), and low (6.6 mm/yr) for SWE. However, comparing variability relative to mean magnitude (Fig. 2b), measured as CV, shows that the vertical, atmospheric-controlled fluxes of P and AET demonstrate lower inter-model variability than the horizontal flux of R or storage components SWE and RZSM. The CV statistic shows how, due to the low overall magnitude of SWE, small differences in estimates have a substantial



effect on magnitude. RZSM, which is often a large overall storage term in the water budget, also shows high CV, indicating that variability in estimates of this component may have the largest impact on hydrologic analyses.

Unfortunately, high-magnitude differences in RZSM (Fig. 3e & 3f) are likely strongly controlled by model-defined soil layer depth and thus limit the utility of disagreement statistics. For example, the TerraClimate and NLDAS2-Noah models apply spatially invariant rootzone soil depth across the model domains (Abatzoglou et al., 2018; Y. Xia et al., 2012a), whereas the NHM-PRMS assumes a variable depth according to vegetative and geophysical parameters (Regan et al., 2018). RZSME estimates are directly controlled by soil depth, returning values of equivalent water depth. RZSMV differences are likely to be less influenced by soil depth due to inherent measurements of fractional volume rather than depth. However, soil-water profiles can change significantly with depth, so attributing model differences strictly by rootzone depth definition is more difficult.

Boxplots of modeled estimates by water budget component (Fig. 3) demonstrate the annual ranges of magnitude generated by various products. Precipitation models (Fig. 3a) exhibit low overall variability, with median magnitudes falling within a distinct 100 mm/yr (700-800 mm/yr) range. The exceptions to this are the CMAP reanalysis datasets, which have a median 647 mm/yr.

Variability among AET estimates (Fig. 3b) is higher, falling within a 200 mm/yr (435-650 mm/yr) range, though attributing differences to model type (e.g. LSMs vs. CMs) is difficult. Generally, CMs (MERRA-2/CLSM, MERRA-Land/CLSM, NHM-PRMS) and conceptual WBMs (NHM-MWBM, GLEAM, TerraClimate) produce greater annual rates of AET as compared to LSMs (NLDAS2-VIC, GLDAS-CLM, NLDAS2-Noah). Exceptions to this are the LSMs NLDAS2-MOSAIC and H-TESSEL that are more similar in annual AET flux to CMs and WBMs. The RS datasets agree more at the CONUS-extent with lower-magnitude estimates.

Estimates of R show three distinct clusters of CONUS magnitudes (Fig. 3c) of 130-190, 220-242, and 310-345 mm/yr. The NLDAS2-Noah and NLDAS2-VIC LSMs, which produced some of the lowest AET rates, fall within the higher magnitude R clusters, as do the JRA-driven SiB models and NHM-PRMS model. The lowest magnitude estimates are the NLDAS2-Mosaic and ERA5-driven H-TESSEL LSMs, grouped with the CM MERRA-Land/CLSM and WBM TerraClimate. LSMs are more likely to estimate greater R than WBMs or CMs.

Boxplots of annually averaged SWE monthly values (Fig. 3d) show that WBMs, notably the NHM-MWMB, generate much higher SWE than most other datasets. Alternatively, there are few discernible patterns between the remaining products, with LSMs and CMs interspersed throughout the gamut of median values. To generalize, the LSMs Noah, GLDAS-CLM, and NLDAS2-Mosaic produce lower-magnitude estimates of monthly SWE. Three different Noah estimates, driven with different meteorological forcings and run both independently (NLDAS2-Noah) or as part of a larger model (e.g. NCEP-





NARR/Eta-Noah), all estimate lower SWE relative to other datasets. The two VIC LSMs show contrasting median monthly SWE magnitude over the CONUS, with the Livneh-VIC median exceeding the NLDAS2-VIC median by more than 200%. The RS AMSR-E/Aqua product agrees more with lower-magnitude estimates at the CONUS extent.

Across all water budget components, most datasets demonstrate lower-magnitude values in the late period (WYs 2000-2014) compared to the early period (WYs 1985-1999). Within the P flux, almost all datasets agree on a decrease in magnitude of hydrologic fluxes from early to late periods. This is similarly reflected in the flux estimates of AET and R, and, to a less uniform extent, the storage components of SWE and RZSM. Several datasets, however, exhibit increased estimates from the early to late periods. For example, the NCEP-DOE reanalysis precipitation, and corresponding runoff derived by forcing the Noah LSM within the Eta atmospheric model, show increased median annual flux rates from the early to late periods.

### 4.1.2 Trends

Only 19 out of 87 component estimates exhibit statistically significant CONUS average domain trends, evaluated with Sen's slope, from 1982 to 2010 (Fig. 4). Within most components, one or more datasets produced a significant slope that is contradicted by one or more datasets. For example, the R estimates from LSMs GLDAS-CLM, JRA-55/SiB, and ERA5/H-TESSEL each show a significant negative trend in annual values across the CONUS over the 1982-2010 period. Conversely, the NCEP-DOE/Eta-Noah LSM shows a significant positive trend over the same period and the remaining datasets, a mix of hydrologic and reanalysis models, show no significant trend. While the NCEP-DOE reanalysis precipitation trend matches that of the NCEP-DOE forced Eta-Noah, model estimates of SWE indicate a significant negative trend.

### 4.2 Ecoregions

### 4.2.1 Magnitude Variability

Variability in modeled hydrologic component estimates in each ecoregion is presented in terms of CV (Fig. 5) and SD (Appendix 1). Ecoregions are organized from west to east within each figure. Results and discussion of variability are focused primarily on CV due to the disparity in annual water flux between regions. By component, P and AET estimates demonstrate lower CV compared to other hydrologic components (Fig. 5a & 5b). P estimates typically range in median CV from 6-12% in the eastern CONUS and 12-21% in the western CONUS. P variability is highest in mountainous regions with more variable topography (e.g. Northwestern Forested Mountains and Temperate Sierras) and lowest in more humid, topographically homogeneous regions (e.g. Eastern Temperate Forests).

The median CVs of AET estimates typically range from 14-21% in the eastern CONUS ecoregions and 19-27% in the western CONUS. Variability is greatest along the Pacific coast and northerly ecoregions, as well as arid and semi-arid ecoregions that are heavily influenced by anthropogenic water use. Of the four largest ecoregions (Table 2), Northwestern Forested Mountains and North American Deserts demonstrate mean CVs of 21% and 19%, respectively.



Estimates of R (Fig. 5c) generally exceed 28% variability throughout the study ecoregions, with median values reaching or exceeding 88% in arid western ecoregions of Mediterranean California, North American Deserts, Southern Semiarid Highlands, and Temperate Sierras. The greatest variability in R estimates is found in snow-sparse arid and semi-arid ecoregions of the western CONUS, as well as eastern regions that are affected by more sporadic precipitation events such as the Tropical Wet Forests region influenced by tropical storm systems. In the large Northwestern Forested Mountains and

Great Plains ecoregions, median dataset variability is 40-57% and 68-72%, respectively.

SWE estimate variability is highlighted for the snow-dominated Northwestern Forested Mountains and Northern Forests ecoregions (Fig. 5d) where annual peak SWE values exceed estimated snowpack in the remaining ecoregions by up to 700% (Appendix 2.2). Of primary interest is the Northwestern Forested Mountains ecoregion that contains the snowmelt dominated regimes of the Rocky, Sierra Nevada, and Cascade mountain ranges that are strong contributors to water supply

for population centers such as Denver, Los Angeles, San Francisco, Portland, and Seattle. In this ecoregion, modeled estimates of monthly mean SWE storage vary by 90-100%, equating to median equivalent water depth standard deviations as high as 40 mm/month.

RZSME estimate variability (Fig. 5e) is greatest, 84-92%, in arid western ecoregions, though other regions fall within a 63-75% CV range. RZSMV shows lower overall variability (Fig. 5f) with less dramatic differences between regions, ranging

from 39-57%, with the greatest variability (44-57%) in central and eastern CONUS ecoregions. As mentioned previously, variability in RZSM estimates, most importantly RZSME, is strongly influenced by model-defined rootzone soil thickness. Spatial differences in RZSME variability measured as CV, therefore, are pronounced in regions with lower water content because CV, a measure of relative variability, is more strongly affected by magnitude differences. Because the range of values is constrained from 0-1 in m3/m3, variability in RZSMV is less influenced by spatial variability in soil water

magnitudes and is therefore a better measure for understanding regional soil water estimate disagreement. Following this logic, RZSMV variability is greater in Tropical Wet Forests, Northern Forests, and Great Plains ecoregions with an average CV of 54%, compared to an average of 44% for the remaining ecoregions.

### 4.2.2 SWE Accumulation and Ablation

Timing of SWE estimates is presented in terms of relative timing (Appendix 2.3 & 2.4). Figures are divided by Northern

Forests and Northwestern Forested Mountains ecoregions, trends of accumulation and ablation, and early and late periods. Negative (positive) values in purple (green), identify models where accumulation or ablation begins more than 1 month earlier (later) than the mean of other datasets. Distributions of relative timing by year are presented in Appendix 2.3 and summarized as percentage of years that are positive or negative in Appendix 2.4. In both ecoregions, variability among datasets is greater in ablation timing than accumulation timing, and overall variability is greater in the Northwestern Forested

Mountains ecoregion than in the Northern Forests.




Relative timing between models can be similar between ecoregions (Appendix 2.3a, 2.4), especially regarding accumulation. For instance, the AMSR-E/Aqua RS and NHM-MWBM CM consistently show later accumulation start dates than other datasets in both ecoregions, while the ERA5-Land/H-TESSEL, JRA-55/SiB, and SNODAS models estimate earlier accumulation dates. Conversely, the JRA-25/SiB and TerraClimate models show different trends in accumulation timing

between regions with earlier SWE accumulation in Northern Forests but later accumulation in Northwestern Forested Mountains.

Regarding ablation, outliers are much more common when comparing models (Appendix 2.3b, 2.4). For example, the DayMET model consistently estimates a much later start of spring ablation while the Eta-Noah models driven with NCEP-DOE and NCEP-NARR reanalyses typically estimate much earlier ablation times, often 1-2 months after the mean

antecedent month (Appendix 2.3). Variability in ablation timing is typically much higher in the Northwestern Forested Mountains ecoregion than in the Northern Forests. The DayMET, ERA5-Land/H-TESSEL, Livneh-VIC, NLDAS2-VIC, and SNODAS models commonly estimate a later start to ablation than the remaining models.

Models that predict earlier (later) accumulation timing and later (earlier) ablation timing represent longer (shorter) periods of growing or available snowpack (Appendix 2.4). The ERA5-Land/H-TESSEL, Livneh-VIC, and SNODAS models estimate

longer snowpack periods than other models in both ecoregions. The AMSR-E/Aqua, JRA-55/SiB, NHM-PRMS, and TerraClimate models estimate longer snowpack periods in only one ecoregion. The NCEP-NARR/Eta-Noah, NLDAS2-Mosaic, and NLDAS2-Noah models estimate shorter snowpack periods in the Northern Forests ecoregion while the GLDAS-CLM, JRA-25/SiB models estimate shorter snowpack periods in the Northwestern Forested Mountains ecoregion.

Attributing relative timing of SWE to model type is difficult at the monthly scale. LSMs estimate both earlier and later

accumulation and ablation antecedences, and the two WBMs in this study show opposing timing trends. Similarly, grouping by organization does not yield significant similarities. For example, the timing of NLDAS2-driven LSMs from the National Aeronautics and Space Administration (NASA) is dissimilar from that of the MERRA-2- and MERRA-Land-driven CLSM, also from NASA. Forcing data also does not yield useful information for identifying controlling variables. Only the NLDAS2-driven models from NASA tend to all show later (earlier) accumulation (ablation) timing. Even the Eta-Noah

LSMs, driven by the NCEP-DOE reanalysis and the corresponding higher-resolution version for North America NCEP-NARR, show different relative timing values in both ecoregions.

### 4.2.3 Trend

Inter-annual trend analyses performed with the Mann-Kendall trend test (τ) over water years 1982-2010 show varying degrees of agreement and disagreement between ecoregions. Figures showing explicit distributions of trends with model

names are supplied in the appendix (Appendix 2.5-2.9). Model disagreement in trend direction is quantified using the unalikeability coefficient (u) (Fig. 6), with 0 and 1 being complete agreement and disagreement, respectively.



Precipitation estimates demonstrate the lowest u, showing no (u = 0) to low (u = 0.13) disagreement in eight ecoregions where datasets show no significant τ (Appendix 2.5). Disagreement is higher in North American Deserts and Temperate Sierras ecoregions (u = 0.41-0.49) where most datasets show a negative τ, and all datasets show a negative τ in the Southern Semiarid Highlands. Coefficient u is higher among AET datasets than P datasets, averaging 0.21 across ecoregions, notably in the North American Deserts and Marine West Coast Forest ecoregions (u = 0.46 and 0.53) where negative and insignificant τs are present (Appendix 2.6). All datasets identify negative AET τ in the Southern Semiarid Highlands and Temperate Sierras. Datasets show positive, negative, and insignificant τ in the Eastern Temperate Forests (u = 0.31) and Northern Forests (u = 0.53). Runoff datasets show the most consistent spatial distribution of u > 0 across the study ecoregions. Disagreement in western ecoregions is caused by conflicting negative and insignificant R τ (Appendix 2.7), with the greatest u in the Southern Semiarid Highlands (u = 0.50) and Temperate Sierras (u = 0.49). Eastern ecoregion τ is generally caused by conflicting positive, negative, and insignificant τs (u = 0.26-44). Ecoregions with good agreement (u = 0-0.35) is generally due to most datasets showing no significant trend.

SWE datasets, limited in this study to the Northwestern Forested Mountains and Northern Forests, show disagreement, u = 0.42 and 0.40, respectively, mostly caused by conflicting negative and insignificant τ, though most datasets show no significant trend (Appendix 2.8). Trend agreement is better among RZSM datasets (mean u = 0.25), though higher u is noted in the Northwestern Forested Mountains, North American Deserts, and Northern Forests, caused by conflicting negative and insignificant τs. Ecoregions with low u typically show no significant τs in RZSM datasets except for the Southern Semiarid Highlands where most datasets show a significant negative τ.

Generally, τ disagreement is highest in North American Deserts and Northern Forests ecoregions and lowest in Mediterranean California, Eastern Temperate Forests, Marine West Coast Forest, and Tropical Wet Forests ecoregions. Trend disagreement is almost always caused by conflicting negative and insignificant trends, indicating that disagreement is due to occurrence and not direction of trend. That is to say, higher u values are caused by disagreement over whether there is or is not a significant trend present in the data, as opposed to higher u values being caused by disagreement between significant negative and significant positive trends.

### 4.2.4 Correlation with Remote Sensing

Spearman's rho (ρ) correlation was calculated for all hydrologic models and reanalysis datasets against remote sensing products with 48 or more months of intersecting temporal extents (Fig. 7, Table 3). Precipitation datasets (Fig. 7a & b), correlated against the GPCP-v3 and TMPA-3B43 products, show high correlation compared against 13 datasets (ρ = 0.93-0.99) with no statistically insignificant values. Variability of P estimate correlations are low (± 0.01-0.09). AET datasets correlated against the MOD16-A2 and SSEBop products (Fig. 7c & d) show poorer correlation in western ecoregions, with mean ρ ranging from 0.59-0.91. Extreme cases are found when correlating against MOD16-A2 in North American Deserts



($\rho$ = -0.28) and SSEBop in Mediterranean California ($\rho$ = 0.25) where four datasets show insignificant $\rho$. The exception in the western CONUS is in Northwestern Forested Mountains where models correlate against both RS datasets well (MOD16-
A2 $\rho$ = 0.91 and SSEBop $\rho$ = 0.89). Correlation against both products is high in eastern ecoregions, ranging from $\rho$ = 0.88-0.96 with standard deviation decreasing as $\rho$ increases.

SWE datasets correlated against the AMSR-E/Aqua product are highlighted for the Northwestern Forested Mountains and Northern Forests ecoregions (Fig. 7e). Modeled estimates correlate better in the Northwestern Forested Mountains ($\rho$ = 0.91) than Northern Forests ($\rho$ = 0.80). RZSM datasets show worse and more variable $\rho$ against remote sensing products
ESA-CCI and SMOS-L4 (Fig. 7f & g) than other components. At the CONUS scale, estimates correlate very poorly against the ESA-CCI product ($\rho$ = 0.17) and only moderately well against SMOS-L4 ($\rho$ = 0.68). Generally, correlation is best in the Mediterranean California and Eastern Temperate Forests ecoregions ($\rho$ = 0.74-0.92). Correlation is worst in Northwestern Forested Mountains, North American Deserts, and Northern Forests (ESA-CCI $\rho$ = 0.44-0.61, SMOS-L4 $\rho$ = 0.12-0.28). Explicit distributions of correlation for each dataset by ecoregion are provided in Appendix 2.10.

**4.2.5 Case Study: Impact of Model Selection on Water Budget Imbalances**

A case study calculating 2,925 10-year water budgets (WY 2001-2010) using 15 P, 15 AET, and 13 R estimates for each of eight ecoregions was performed to demonstrate quantitatively how model selection can affect research results. Each water budget was solved for a relative imbalance ($R\varepsilon$) with Eq. (7), estimating model error as a fraction of total water flux in P.

Histograms of results overlain with boxplots (Fig. 8) demonstrate the range and distribution of potential water budget $R\varepsilon$s.
In most ecoregions, $R\varepsilon$s exhibit an approximately normal distribution. Four ecoregions dominating the area of the CONUS (Fig. 8a-d), constituting 90% of the CONUS area (Table 2), are the focus of these results: the western ecoregions Northwestern Forested Mountains and North American Deserts, and the eastern ecoregions Great Plains and Eastern Temperate Forests (11%, 19%, 29%, and 31% of CONUS area, respectively).

Of these larger domains, water budget $R\varepsilon$'s in the eastern ecoregions show much lower variability (Table 4), with Great
Plains and Eastern Temperate Forests yielding SDs of 18.2 percentage points and 14.3 percentage points, respectively, and medians of 3.9% and -0.4%, respectively. Similarly, the 10th and 90th percentiles (P10 and P90) are -23.2% and 23.6% for Great Plains, and -19.9% and 17.2% for Eastern Temperate Forests. In contrast, the major western ecoregions exhibit much higher variability, with Northwestern Forested Mountains and North American Desert SDs of 45.6% and 27.0%, respectively, and medians of -6.7% and 4.7%, respectively. P10 and P90 are higher than eastern regions as well, yielding -
83.3% and 23.8% for Northwestern Forested Mountains and -33.2% and 36.6% for North American Deserts. Smaller ecoregions (Fig. 8e-h) yield similar spatial trends in variability with the eastern Northern Forests ecoregion showing lower SD and P10/90 than the western Marine West Coast Forest and Mediterranean California ecoregions.



The Northwestern Forested Mountains region, accounting for 11% of the CONUS area, is unique among ecoregions in this study. It exhibits the greatest magnitude variability in iterative water budget Rε, P10 - P90 Rε ranges, skewness (-1.6), and

kurtosis (3.7). Furthermore, the difference between the ensemble water budget Rε and median iterative water budget Rε is greater than any other ecoregion (-6.3 points).

## 5 Discussion

Results of model comparisons demonstrate the effect that model disagreement can have on both regional- and continental-scale research. Comparisons of P estimates agree with previous findings (Derin and Yilmaz, 2014; Guirguis and Avissar,

2008; Sun et al., 2018), noting increased variability between products in regions of complex topography. AET comparisons showed that LSMs are more likely to produce lower annual AET than CMs and WBMs, similar to results found by Mueller et al. (2011), and that model disagreement is higher in Pacific regions, similar to the comparison and validation of MOD16-A2, NLDAS2-Noah, and ALEXI by Carter et al. (2018). However, this paper finds some differences from the work of Carter et al. (2018), noting higher variability of AET estimates in arid, semi-arid, and mountainous regions, as well as lower

variability in eastern regions. Haddeland et al. (2011) found that LSMs underestimate R partitioning relative to global hydrologic models (not used in this paper). Here, we find that LSMs are more likely to estimate greater R than either WBMs or CMs. Xia et al. (2012b, 2012a) noted greater model disagreement in R estimates between several LSMs in northeastern and western mountainous regions of the CONUS. This paper, using CV as a measure of relative variability, alternatively shows that modeled runoff variability is higher in the arid and semi-arid regions of the western CONUS where annual R

rates are lower.

Previous analyses of modeled SWE estimates focused largely on validation of winter snowpack magnitudes against the SNODAS model or on the timing of accumulation and ablation periods. Broxton et al. (2016) found that LSMs estimated earlier ablation timing than observational measurements, contrasting the results of Essery et al. (2009) and Rutter et al. (2009) who noted later timing in LSMs. Our results indicate the alternative argument that neither snow accumulation nor

ablation timing can be accurately attributed solely to model design, noting differences in relative timing of as much as two months between LSMs. Contrasting the arguments by Murdryk et al. (2015) and Broxton et al. (2016), who concluded that variability in estimates may be controlled by differences in model structure, results here show that attributing model differences to structure is likely impossible without controlling for forcing data, model parameters, and calibration methods. Properly identifying controls on model differences requires robust MIPs wherein hydrologic models are operated within the

confines of strict input data and calibration schemes, such as those performed by Haddeland et al. (2011), Kollet et al. (2017), or Rosenzweig et al. (2013).

Perhaps the most important variable controlling differences in RZSME are the discrepancies in model-defined rootzone depth. Typically the largest surface storage component in the hydrologic cycle, differences in RZSME estimates will more





greatly affect water availability calculations than any other component. Koster et al. (2009) argued that modeled soil
moisture, related in units of equivalent water depth (i.e. RZSME), should not be considered direct measures of actual soil
water content but instead should be used as relative values to identify seasonal to annual trends and responses to changing
climatology. However, LSM estimates of soil moisture depth are commonly applied in research applications as direct
measures of soil water content. For example, groundwater storage trends are calculated from Gravity Recovery and Climate
Experiment (GRACE) solutions of terrestrial water storage anomalies (Rodell et al., 2007b; Scanlon et al., 2012; Thomas
and Famiglietti, 2019; Voss et al., 2013). Groundwater storage is calculated as the remainder after removing surface storage
components such as modeled surface water storage, SWE, and RZSME storage values from GRACE terrestrial water
storage. Because estimates of RZSME can vary by 63-92%, groundwater storage values derived from GRACE solutions will
be significantly affected. Groundwater storage values are most commonly extracted using the NLDAS2- or GLDAS-driven
LSMs Mosaic, Noah, VIC, or CLM that agree more in monthly RZSME magnitude than with other estimates from WBMs or
CMs. However, even those products show differences in RZSME estimates of up to 200 mm per year. Therefore,
quantifying differences in RZSME is useful in understanding uncertainty propagation in research results.

In terms of evaluating surface water availability, snowmelt from the Northwestern Forested Mountains ecoregion is a
primary supply of water for many western population centers. However, disagreement between models of annual snowpack
in this study, measured as SWE, averages 97% relative variability and timing on accumulation or ablation can vary by up to
two months. These levels of disagreement strongly affect the accuracy of predictive and retrospective snow water analyses
of both present snowpack and long-term trends.

LSMs, typically targeted to estimate the vertical AET flux, generally overestimate R compared to CMs. Conversely, CMs,
targeted to estimate the horizontal R flux, often overestimate AET compared to LSMs. This is likely an example of
equifinality (Beven, 2006), wherein target variable accuracy may be reached without accurately representing the complete
hydrologic system. In comparing model output by organization, estimates of P, AET, and SWE produced by NASA are
often lower in magnitude than those produced by the USGS and European Centre for Medium-Range Weather Forecasts
(ECMWF), indicating organizational differences in model operation methods, such as forcing data selection or calibration
methodology.

Results of the water budget case studies demonstrate that water budget Rεs are more variable in the western CONUS, with
SDs of ±27.0-45.6% by ecoregion, although even eastern CONUS Rεs range in variability from ±14.3-18.9% by ecoregion.
In all regions, various combinations of products can result in both positive and negative imbalances (ε), yielding alarmingly
different research implications. Positive ε (P > AET + R) means that more water is entering a system than leaving.
Assuming limited influence of model uncertainty, this would imply that excluded natural or anthropogenic fluxes are
removing water from the P = AET + R system. This could be interpreted as the presence of long-term natural contributions
to storage components, such as soil moisture or groundwater, or as the presence of anthropogenic extractions. The same





region, depending on models selected, could yield negative ε (P < AET + R), implying the presence of additive fluxes, such as releases from surface storage or increased irrigation from imported water.

Results in this paper supplement the work of Haddeland et al. (2011) who argued that climate change effect modeling on large-scale hydrologic processes should utilize a range of model estimates rather than rely on a single model realization.

Indeed, we show that model variability of storage components often exceeds the magnitude of the measurements themselves, calling into question the functional application of terrestrial storage estimates. Our work shows that ensembles of model estimates are not only useful, but are in fact a necessity, to best understand both terrestrial hydrology and quantify our degree of knowledge. Improved ensembling methods, such as was done by Zaherpour et al. (2019), will likely be invaluable resources in the fields of terrestrial hydrology, climatology, and meteorology modeling.

This comprehensive and systematic comparison helps elucidate where we as a scientific community could better target research efforts to reconcile large-extent estimates of the hydrologic cycle.  Most importantly, differences in modeled RZSME, both in terms of magnitude depth and correlation against RS products, would need to be addressed at the continental-scale to better understand the largest surface store. Further, estimates in SWE, AET, and R need to be reconciled in much of the western CONUS to provide a more accurate retrospective and operational overview of the hydrologic cycle.

Finally, differences in hydrologic estimates warrant incorporation into future analysis that use these products to provide the scientific community with a more robust understanding of results confidence and areas of uncertainty.  All datasets explored in this paper are easily obtained from publicly available sources, and modern data processing workflows are applied to multiple datasets without difficulty.

## 6 Conclusion

Model selection can significantly alter results and findings when used as a substitute for observational data.  Publicly available modeled estimates are commonly used within the scientific community, though literature effectively evaluating differences in magnitude and trend in the context of application is rare.  Model Intercomparison Projects abound within the scientific literature, often focusing on the causes of model disagreement (e.g., forcing data, model structure, calibration) and validating against in-situ observational data despite limited availability.  In contrast, this study investigated how publicly

available modeled estimates of hydrologic flux and storage components can affect scientific analyses by quantifying disagreement between numerous hydrologic models, reanalysis datasets, and remote sensing products.

Results show that flux and storage magnitudes disagree most greatly in the western CONUS, with CVs of precipitation (P) 11-22%, actual evapotranspiration (AET) 14-27%, runoff (R) 28-153%, snow water equivalent (SWE) 92-102%, and rootzone soil moisture (RZSM) 39-92%.  In the eastern CONUS, disagreement is somewhat lower with CV of P 5-15%,

AET 13-23%, R 29-96%, SWE 64-70%, and RZSM 44-81%.  Inter-annual trends in estimates from 1982-2010 show more comprehensive agreement for P and AET fluxes, but common disagreement for R, SWE, and RZSM. Disagreement in trends



(i.e. positive versus negative versus insignificant) between models is typically a result of conflicting negative and insignificant trends rather than between negative and positive trends, indicating that disagreement is due to occurrence and not direction of trend. Correlating fluxes and stores against remote sensing-derived products shows poor overall correlation
in the western CONUS for AET and RZSM. P correlates well in all regions, and SWE correlates well in the primary regions of Northwestern Forested Mountains and Northern Forests.

A water budget analysis, performed by iterating through all combinations of publicly available modeled products highlighted in this research, shows that in large eastern ecoregions model selection can result in relative imbalances ranging from -50 to 50%. In larger western ecoregions, relative imbalances can range from -150 to 60%.

Publicly available modeled estimates of the hydrologic system accelerate the development of scientific research by reducing the necessary workload, processing time, and technical expertise for researchers. In addition, estimates fill knowledge gaps in fluxes and stores where observational data are incomplete. Metrics of disagreement in component estimates presented here help to describe the uncertainty of the scientific community's current state of knowledge. The uncertainty inherent in modern datasets can affect results of studies as diverse as satellite-derived groundwater estimates and predictive snowmelt
analyses. Our results highlight problem areas within CONUS-extent hydrologic estimation efforts that warrant better understanding and addressing in future endeavors. Most importantly, our results show that the most important issues to reconcile are disagreement, in terms of both magnitude and long-term trend, of (a) RZSM storage across the CONUS, (b) SWE storage in western mountains, and (c) hydrologic fluxes in western arid and semi-arid regions. Additionally, our results support the findings of previous studies and agree that future work applying modeled data would prove to be more
accurate, more informative, and more robust by incorporating model ensembles to provide confidence intervals, better quantify result uncertainty, and improve overall accuracy.

**Acknowledgements:** This work was performed under the U.S. Geological Survey's Water Availability and Use Science Program and the Hydrologic Sciences and Engineering fellowship from the Colorado School of Mines. The authors are grateful for the early assistance of Flannery Dolan (Tufts University), Shirley Leung (University of Washington), and Adam
Price (University of California Santa Cruz). All the data used in this study were obtained from public domains and are freely available. Data used for this study were converted from their primary structures (rasters and shapefiles) to EPA Ecoregions Level 1 and are available in Saxe et al. (2020). Any use of trade, firm, or product names is for descriptive purposes only and does not imply endorsement by the U.S. Government.

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





## Appendix 1

| Acronym | Product Description |
| --- | --- |

**Hydrologic Models**

| CPC | Climate Prediction Center |
| --- | --- |
| | *Rootzone soil moisture (equivalent water depth) estimated through a one-layer water balance model forced by CPC reanalysis precipitation and temperature. Resolutions are monthly at 1/2° from 1948 to present for the globe* (Fan and van den Dool, 2004). *Author's acknowledgement: CPC soil moisture data provided by the NOAA/OAR/ESRL PSD, Boulder, Colorado, USA, from their Web site at https://www.esrl.noaa.gov/psd/.* |
| CSIRO-PML | Commonwealth Scientific and Industrial Research Organisation |
| | *Estimates of evapotranspiration generated by a spatially explicit Penman-Monteith-Leuning model constrained annually by the Fu hydroclimatic model. The model is driven by precipitation, temperature, vapour pressure, windspeed, and radiation estimates. Datasets used include reanalysis climatology and meteorology products and ex-situ surface and vegetation data. Resolutions are monthly at 1/2° spatial resolution from 1981-2012 for the globe* (Zhang et al., 2016). |
| ERA5/H-TESSEL | Hydrology revised-Tiled ECMWF Scheme for Surface Exchanges over Land forced with the ERA5 reanalysis |
| | *From the European Centre for Medium-Range Weather Forecasts (ECMWF), a land surface model, improved from the original TESSEL model, estimates land surface and subsurface fluxes and stores. The modeled is structured with four soil layers (0-7, 7-28, 28-100, 100-289 cm), a single snow layer, two sub-grid vegetation types, and spatially variant soil type. This paper uses model output forced with ERA5 reanalysis data (see below), provided by C3S in conjunction with the reanalysis product. Resolutions are hourly at 0.25° from 1979 to present for the globe* (Balsamo et al., 2009; C3S, 2017). *Author's acknowledgement: Contains modified Copernicus Climate Change Service Information [2019].* |
| ERA5-Land/H-TESSEL | Hydrology revised-Tiled ECMWF Scheme for Surface Exchanges over Land forced with the ERA5-Land reanalysis |
| | *From the ECMWF, a land surface model, improved from the original TESSEL model,* |





*estimates land surface and subsurface fluxes and stores. See above for a description of model structure. This paper uses model output forced with ERA5-Land reanalysis data (see below), provided by C3S in conjunction with the reanalysis product. Resolutions are hourly at 0.1° from 2001 to present for the globe (Balsamo et al., 2009; C3S, 2019). Author's acknowledgement: Contains modified Copernicus Climate Change Service Information [2019].*

GLDAS-CLM     Community Land Model, V2 driven by the NASA GLDAS

*From the National Aeronautics and Space Administration (NASA) Global Land Data Assimilation System (GLDAS), a single-column land surface model that estimates surface and subsurface fluxes and stores within independent spatial domains. The model is structured with variable layer spacing for soil temperature and moisture, multilayer snow process parameterization, TOPMODEL-concept runoff parameterization, a canopy photosynthesis-conductance model, and tiling treatments for sub-grid energy and water balances. Model input requirements are land surface types, soil and vegetation parameters, model initialization states, and climatological forcing data. This study uses the operational version driven by the NASA GLDAS. Resolutions are sub-daily at 1° from 1979 to present for the globe (Dai et al., 2003; Rodell et al., 2007a, 2004).*

GLEAM     Global Land Evaporation Amsterdam Model

*Estimates land evaporation, surface soil moisture, rootzone soil moisture, potential evaporation, and evaporative stress conditions. Potential evapotranspiration is calculated using the Priestly-Taylor equation on reanalysis, in-situ, and ex-situ meteorological measurements. Actual evapotranspiration is calculated using an evaporative stress factor based on rootzone soil moisture estimates and microwave ex-situ observations. Rootzone soil moisture is estimated through a water balance model. Model generation 3.3 replaces reanalysis measurements from ERA-Interim with ERA-5 (see below). Model version 3.3b differs from 3.3a by utilizing primarily ex-situ data, excluding reanalysis products. Resolutions are daily at 1/4° from 1980 to present for the globe (Martens et al., 2017; Miralles et al., 2011).*

JRA-25/SiB     Simple Biosphere model forced with Japanese 25-year Reanalysis

*Land surface flux and store estimates generated as output from the Japan Meteorological*



*Agency (JMA)-operated Simple Biosphere model (SiB) forced with Japanese 25-year ReAnalysis (JRA-25) reanalysis. The predecessor to modern land surface models such as Mosaic, the model is biophysically based and structured with two vegetation layers (canopy and ground cover), and three soil layers. Resolutions are sub-daily to daily at T106 (~110 km) from 1979 to 2004 for the globe (NCAR Research Staff (Eds), 2016; Onogi et al., 2007; Sellers et al., 1986).*

JRA-55/SiB
Simple Biosphere model forced with Japanese 55-year Reanalysis
*Land surface flux and store estimates generated as output from the JMA-operated SiB model forced with Japanese 55-year ReAnalysis (JRA-55) reanalysis. See above for model description. Resolutions are sub-daily to daily at T319 (~55 km) from 1957 to present for the globe (Kobayashi et al., 2015; Kobayashi and NCAR Research Staff (Eds), 2019; Sellers et al., 1986).*

Livneh-VIC
Variable Infiltration Capacity (VIC) model forced and calibrated by Livneh et al. 2013
*Modeled estimates of soil moisture, snow water equivalent, discharge, and surface heat fluxes generated by forcing the VIC model with the Livneh et al. 2013 reanalysis meteorological dataset (see below). See below (NLDAS2-VIC) for description of model structure. Resolutions are sub-daily to daily at 1/16° from 1915 to 2011 for the conterminous United States (Liang et al., 1994; Livneh et al., 2013).*

MERRA-Land/CLSM
Catchment Land Surface Model forced with MERRA-Land reanalysis
*A land surface model developed to improve treatment of horizontal hydrologic process in response to previous land surface model over-attention to vertical processes. The model is structured around tiled hydrologic catchments defined by topography, two soil layers, and three snow layers. This paper uses surface estimates generated from the MERRA-Land-forced version from GEOS-5, released in conjunction with the MERRA-Land reanalysis product (see below). Resolutions are sub-daily at 1/2° from 1980 to 2016 for the globe (Koster et al., 2000; Reichle et al., 2011; Rienecker et al., 2011).*

MERRA-2/CLSM
Catchment Land Surface Model forced with MERRA-2 reanalysis
*A land surface model estimating surface and subsurface hydrologic fluxes and stores. See above for model description. This paper uses estimates generated from the MERRA-2-forced*



*version from GEOS-5, released in conjunction with the MERRA-2 reanalysis product (see below). Resolutions are sub-daily at 1/2° from 1980 to present (Gelaro et al., 2017; Reichle et al., 2017).*

NCEP-DOE/Eta-Noah     Noah land surface model forced with National Centers for Environmental Prediction-Department of Energy, R-2 reanalysis

*Noah land surface model estimates surface and subsurface hydrologic fluxes. See below (NLDAS2-Noah) for model description. Model estimates here are generated through the Noah land surface model component of the NCEP Eta atmospheric model forced with NCEP-DOE reanalysis (see below), released in conjunction with the NCEP-DOE product. Resolutions are sub-daily at T62 (~210 km) gridding from 1979 to present for the globe (Kalnay et al., 1996; Kanamitsu et al., 2002). Author acknowledgment: NCEP_Reanalysis 2 data provided by the NOAA/OAR/ESRL PSD, Boulder, Colorado, USA, from their Web site at https://www.esrl.noaa.gov/psd/.*

NCEP-NARR/Eta-Noah     Noah land surface model forced with National Centers for Environmental Prediction-North American Regional Reanalysis

*Noah land surface model estimates produced as a component of the larger Eta atmospheric model. See below (NLDAS2-Noah) for model description. Model estimates here are generated through the Noah surface model component of the NCEP Eta atmospheric model forced with NCEP North American Regional Reanalysis (NCEP-NARR), released in conjunction with the NCEP-NARR product. Resolutions are sub-daily at 32 km from 1979 to present for North America (Mesinger et al., 2006). Author acknowledgement: NCEP Reanalysis data provided by the NOAA/OAR/ESRL PSD, Boulder, Colorado, USA, from their Web site at https://www.esrl.noaa.gov/psd/.*

NHM-MWBM     National Hydrologic Model-framework Monthly Water Balance Model

*A water balance model applied within the U.S. Geological Survey's National Hydrologic Model framework that utilizes a monthly accounting procedure to estimate evapotranspiration, runoff, soil moisture, and snow water equivalent based on methodology from Thornthwaite (1948). Resolutions are monthly at Hydrologic Response Units from 1949 to 2010 for the conterminous United States (McCabe and Markstrom, 2007).*





| | |
|---|---|
| NHM-PRMS | National Hydrologic Model-framework Precipitation Runoff Modeling System |
| | *A process-based, deterministic hydrologic model applied within the framework of the U.S. Geological Survey's National Hydrologic Model that estimates various surface and subsurface fluxes and stores using a conceptualized watershed composed of a series of reservoirs, stream segments, and lakes maintained with a balanced water budget. The model utilizes reanalysis and ex-situ physical characteristic data of topography, soils, vegetation, geology, and land use to derive required parameters, and is driven by precipitation and temperature reanalysis products. Resolutions are daily at Hydrologic Response Units from 1980 to 2016 for the conterminous United States (Markstrom et al., 2015; Regan et al., 2018).* |
| NLDAS2-Mosaic | Mosaic model driven by the NASA NLDAS, Phase 2 |
| | *A land surface model, directly descended from the SiB land surface model, that estimates surface and subsurface fluxes and stores, originally targeted to be coupled with climate and weather models. The model is structured to allow vegetation control over surface energy and water balances, a canopy interception reservoir, three soil reservoirs (thin surface, middle "rootzone", and lower "recharge"), and a complete snow budget. The model name is derived from the mosaic approach of tiling sub-grid cells into homogeneous vegetation types with independent energy balances. This study uses the operational version driven by the NASA NLDAS Phase 2 (see below), wherein the model was configured to support up to 10 tiles per grid cell, each with specified predominant soil types and three spatially invariant soil layers. Resolutions are hourly at 1/8° from 1979 to present for North America (Koster and Suarez, 1996; Youlong Xia et al., 2012b; Y. Xia et al., 2012b).* |
| NLDAS2-Noah | Noah model driven by the NASA NLDAS, Phase 2 |
| | *A grid-based land surface model, simpler than the Noah-MP version, originally developed as a component of the NOAA-NCEP Eta model and updated for use in the NASA NLDAS, Phase 2 (see below), and is used within the Weather Research and Forecasting (WRF) atmospheric model, the National Oceanic and Atmospheric Administration-NCEP Climate Forecast System (NOAA-NCEP CFS), and the NOAA Global Forecast System. The model is structured with four spatially invariant thickness soil layers (three layers forming the rootzone system in non-forested regions and four in forested regions) and can simulate soil freeze-thaw processes. This study uses the operational version (Noah-2.8) driven by the NASA NLDAS, Phase 2 (see below). Resolutions are hourly at 1/8° from 1979 to present for North America* |



(Ek et al., 2003; Youlong Xia et al., 2012b; Y. Xia et al., 2012a).

NLDAS2-VIC     Variable Infiltration Capacity model driven by the NLDAS, Phase 2

*A semi-distributed, grid-based hydrologic model estimating surface and subsurface fluxes and stores. Model structure is composed of three soil layers (top spatially invariant 10 cm thickness, others spatially-variant) with rootzone depth controlled by vegetation. The model utilizes sub-grid vegetation tiling similar to the Mosaic land surface model (see above), and a two-layer energy balance snow model. This study uses the operational version driven by the NASA NLDAS, Phase 2 (see below). Resolutions are hourly at 1/8° from 1979 to present for North America (Liang et al., 1994; Youlong Xia et al., 2012b; Y. Xia et al., 2012c).*

SNODAS     SNOw Data Assimilation System

*Estimates snow cover and snow water equivalent by integrating modeled snow estimates with observational and reanalysis data from in-situ and ex-situ sources. The primary estimation method is a physically based, spatially-distributed energy- and mass-balance snow model forced by downscaled output from the National Weather Service Rapid Refresh weather forecast model. In-situ and ex-situ data sources are applied depending on difference fields between modeled and observed values, and used to perform immediate model calibrations. Resolutions are daily at 1 km from 2003 to present for North America (Barrett, 2003; National Operational Hydrologic Remote Sensing Center, 2004).*

TerraClimate     TerraClimate

*A dataset utilizing both reanalysis and water budget modeling to generate estimates of climate, meteorological, and hydrologic variables. Climate and meteorological estimates are calculated by applying interpolated time-varying anomalies from both CRU Ts4.0 and JRA-55 to the higher resolution WorldClim climatology. Hydrologic estimates of evapotranspiration, precipitation, temperature, and soil water capacity are generated through a modified Thornthwaite-Mather climatic water balance model. Water balance-derived resolutions are monthly at 1/2° from 1958 to present for the globe (Abatzoglou et al., 2018).*

VegET     Vegetation ET

*Estimates rootzone soil moisture and evapotranspiration through a precipitation-driven one-*





*dimensional rootzone water balance model. Because the model is parameterized to operate on a control volume using water holding capacity, it captures only evapotranspiration from vegetation sources (i.e. natural conditions) and does not consider anthropogenic (i.e. non-natural) water use. Data provided via personal correspondence with the authors. Resolutions are daily at 1 km from 2000 to 2014 for the conterminous United States* (Senay, 2008; Velpuri and Senay, 2017).

| **Reanalysis** | |
|---|---|
| CanSISE | Canadian Sea Ice and Snow Evolution Network, V2 |

*Estimates snow water equivalent (SWE) by merging five observation-based estimates through an adapted ensemble mean methodology. Merged products are (1) GlobSnow-combined SWE (merges ex-situ passive microwave and in-situ weather station observations), (2) ERA-Interim reanalysis, (3) MERRA reanalysis, (4) SWE from the Interaction Sol-Biosphère-Atmosphère (ISBA) land surface model forced with ERA-Interim reanalysis (see below), and (5) NASA Global Land Data Assimilation System reanalysis. Resolutions are daily at 1° from 1981 to 2010 for the Northern Hemisphere* (Mudryk et al., 2015; Mudryk and Derksen, 2017).

| CMAP | CPC Merged Analysis of Precipitation |
|---|---|

*From the CPC, estimates precipitation by blending various in-situ, ex-situ, and reanalysis datasets. The "Standard" product estimates precipitation by blending ex-situ Global Precipitation Index (GPI), OLR Precipitation Index (OPI), Special Sensor Microwave/Imager (SSM/I) emission, and Microwave Sounding Unit (MSU) data with in-situ precipitation gauge measurement data. The "Enhanced" product additionally includes blended NCEP-NCAR reanalysis (see below) precipitation estimates. Resolutions are monthly at 2 1/2° from 1979 to present for the globe* (Xie and Arkin, 1997). *Author's acknowledgement: CMAP Precipitation data provided by the NOAA/OAR/ESRL PSD, Boulder, Colorado, USA, from their Web site at https://www.esrl.noaa.gov/psd/.*

| DayMET | Daily Surface Weather Data on a 1-km Grid for North America, V3 |
|---|---|

*Version 3 of the DayMET model, generates estimates of temperature, precipitation, radiation, vapor pressure, snow water equivalent, and day length from in-situ observations of temperature and precipitation and digital elevation data. Interpolation is performed using the*



*spatial convolution of a truncated Gaussian weighting filter with an iterative station density algorithm applied to heterogeneous observation distribution in complex terrain. Snow water equivalent is estimated using a simple temperature-based snowmelt model from Running and Coughlan (1988). This paper assigns DayMET snow water equivalent estimates to the "Reanalysis" category because of the snow model's simplicity and lack of a physical or conceptual hydrologic framework. Descriptions of the current model framework are best described at the ORNL DAAC website (daac.ornl.gov/DAYMET/guides/Daymet_V3_CFMosaics.html). Resolutions are daily at 1 km from 1980 to present for North America (Thornton et al., 2017, 2000, 1997).*

ERA5       ECMWF ReAnalysis, Fifth Product

*Generates numerous estimates of climate and meteorological variables by assimilating observational data from 55 ex-situ and 19 in-situ sources using the 4D-Var variational method. Previous generations of the ERA product were FGGE, ERA-15, ERA-40, and ERA-Interim, all now out of service. Resolutions are hourly at 0.25° from 1979 to present for the globe (C3S, 2017). Author's acknowledgement: Contains modified Copernicus Climate Change Service Information [2019].*

ERA5-Land       ECMWF Reanalysis, Fifth Product

*Estimates of climate, meteorological, and surface hydrology fluxes by replaying the ERA5 reanalysis (see above) land component at higher spatial resolution. Resolutions are hourly at 0.1° from 2001 to present for the globe (Balsamo et al., 2009; C3S, 2019). Author's acknowledgement: Contains modified Copernicus Climate Change Service Information [2019].*

GPCC       Global Precipitation Climatology Centre

*Monthly accumulated precipitation estimates based on 67,200 in-situ observational stations with decade or longer temporal spans. The dataset used in this paper is V7, one of two GPCC Full Data Products, which the product's authors suggest are the highest accuracy of their datasets. Resolutions are monthly at 1/2° from 1901 to 2013 for the globe (Becker et al., 2013; Schneider et al., 2011). Author's acknowledgement: GPCC Precipitation data provided by the NOAA/OAR/ESRL PSD, Boulder, Colorado, USA, from their Web site at https://www.esrl.noaa.gov/psd/*



| gridMET | gridMET (or, METDATA) |
|---|---|
| | *Estimates temperature, precipitation, downward shortwave radiation, wind-velocity, humidity, relative humidity, and specific humidity by blending PRISM reanalysis (see below) climate data with NLDAS2 reanalysis (see below) data. Resolutions are daily at 1/24° from 1979 to present for the conterminous United States* (Abatzoglou, 2013). |
| Livneh et al. 2013 | Livneh daily CONUS near-surface gridded observed meteorological data |
| | *Near-surface meteorological estimates generated from approximately 20,000 NOAA Cooperative Observer station daily datasets. Precipitation and temperature in-situ observations are converted to grids using the synergraphic mapping system, wind data linearly interpolated from the lower resolution NCEP-NCAR reanalysis dataset. Other variables were derived using methods from the mountain microclimate simulator. Daily temperature data is converted to 3-hourly using spline interpolation. Resolutions are daily at 1/16° from 1915 to 2011 for the conterminous United States* (Livneh et al., 2013). *Author acknowledgement: Livneh data provided by the NOAA/OAR/ESRL PSD, Boulder, Colorado, USA, from their Web site at [https://www.esrl.noaa.gov/psd/](https://www.esrl.noaa.gov/psd/).* |
| Maurer et al 2002 | Maurer et al. 2002 |
| | *Meteorological and surface hydrologic estimates generated through both reanalysis and hydrologic modeling. Precipitation gridded from in-situ daily measurement stations using the synergraphic mapping system algorithm scaled to match PRISM (see below) long-term averages. Surface estimates derived through the VIC (see above) hydrologic model. Resolutions are daily at 1/8° from 1950-1999 for North America* (Maurer et al., 2002). |
| MERRA-Land | Modern-Era Retrospective analysis for Research and Applications with improved land surface variables |
| | *The predecessor to MERRA-2 (see below) estimates meteorological and surface hydrologic components. Meteorological estimates are derived through the GEOS-5 atmospheric general circulation model by assimilating in-situ (3DVAR analysis algorithm) and ex-situ (Community Radiative Transfer Model) observational data. This paper uses the Land Surface Diagnostics precipitation. Resolutions are sub-daily at 1/2° from 1980 to 2016 for the globe* (Rienecker et al., 2011). |





| MERRA-2 | Modern-Era Retrospective analysis for Research and Applications, V2 |
|---|---|

*The most recent version of the MERRA product differs from MERRA-land by including updates to models, algorithms, observing systems, and ex-situ processing methods, as well as 14 additional ex-situ data sources. Precipitation is derived from global precipitation products disaggregated to hourly time steps using MERRA-Land precipitation. Precipitation estimates used in this paper are from the Land Surface Diagnostics category. Resolutions are sub-daily at 1/2° from 1980 to present (Gelaro et al., 2017).*

| NCEP-DOE | National Centers for Environmental Prediction-Department of Energy, R-2 |
|---|---|

*The updated (R-2) version of the NCEP-NCAR reanalysis, estimates climate, meteorological, and surface hydrologic components. Reanalysis estimates of atmospheric variables are performed through assimilation and modeling of in-situ and ex-situ observational data. Resolutions are sub-daily at T62 (~210 km) gridding from 1979 to present for the globe (Kalnay et al., 1996; Kanamitsu et al., 2002). Author acknowledgment: NCEP-DOE data provided by the NOAA/OAR/ESRL PSD, Boulder, Colorado, USA, from their Web site at https://www.esrl.noaa.gov/psd/.*

| NLDAS2 | North American Land Data Assimilation System, Phase 2 |
|---|---|

*Compiled data used to drive land surface models (see above), including (1) land surface parameters of vegetation, soil, topography, and temperature; (2) surface forcing fields of precipitation, radiation, temperature, humidity, wind, and pressure derived by blending various in-situ and ex-situ datasets. Precipitation, specifically, is derived by blending temporally disaggregated reanalysis (CPC and NARR, see above) and ex-situ (Doppler Stage II and CMORPH) estimates. Resolutions are hourly at 1/8° from 1979 to present for North America (Youlong Xia et al., 2012b; Xia, NCEP/EMC, et al., 2009).*

| PRISM | Parameter-elevation Regressions on Independent Slopes Model |
|---|---|

*Reanalysis estimates of precipitation and temperature calculated using a climate-elevation regression model, utilizing information of location, elevation, coastal proximity, and other geophysical parameters. Resolutions are daily at 800 m (paid) and 4 km (free) from 1895 to present for the United States (PRISM Climate Group, OSU, 2004).*

| Reitz et al 2017 | Reitz et al. 2017 |
|---|---|



*Reanalysis estimates of groundwater recharge, quick-flow runoff, and evapotranspiration. Runoff and evapotranspiration are calculated using regression equations derived from observational water balance data using land cover, temperature, and precipitation information. Groundwater recharge is calculated as the remainder of a water balance using runoff, evapotranspiration, and precipitation. Resolutions are annual (monthly provided via personal correspondence with authors) at 800 m from 2000 to 2013 for the conterminous United States* (Reitz et al., 2017).

UoD-v5                 University of Delaware Air Temperature and Precipitation, V5

*Reanalysis precipitation and temperature estimates calculated from in-situ observational data with an interpolation algorithm based on the spherical Shepard's distance-weighting method with Digital Elevation Model information. This paper uses the fifth product version. Resolutions are monthly at 1/2° from 1950 to 1999* (Willmott and Matsuura, 2001). *Author acknowledgement: UoD precipitation data provided by the NOAA/OAR/ESRL PSD, Boulder, Colorado, USA, from their Web site at [https://www.esrl.noaa.gov/psd/](https://www.esrl.noaa.gov/psd/).*

WaterWatch             WaterWatch

*Estimates of runoff derived from the U.S. Geological Survey's in-situ streamgauge network. Resolutions are monthly at watershed, Hydrologic Unit Levels 2-8, and state polygons from 1901 to present* (Jian et al., 2008).

**Remote Sensing**

AMSR-E/Aqua            AMSR-E/Aqua Monthly L3 Global Snow Water Equivalent EASE-Grids, V2

*Scientifically identical to Version 1 (updates to product maturity code in V2), estimates SWE using passive microwave data collected from the Advanced Microwave Scanning Radiometer – Earth Observing System (AMSR-E) instrument hosted on NASA's Aqua satellite. Microwave measurements are converted to SWE estimates using the AMSR-E Snow Water Equivalent Algorithm that utilizes brightness temperature differences calculated using the dense media radiative transfer equation and pre-selected snowpack profiles to develop an artificial neural network. Probable SWE estimate ranges are then restricted using surface temperature and land cover attributes derived from the ex-situ MODIS sensor hosted on the NASA Terra satellite. Resolutions are daily at 25 km from 2002 to 2011 for the globe* (Chang et al., 2003;



Chang and Rango, 2000; Tedesco et al., 2004).

ESA-CCI          European Space Agency – Climate Change Initiative

*Volumetric soil moisture estimates generated by merging both active and passive ex-situ microwave data-derived soil moisture products into three products: ACTIVE, PASSIVE, and COMBINED. Passive microwave products are from the Scanning Multichannel Microwave Radiometer (SMMR), SSM/I, Tropical Microwave Imager (TMI), Advanced Microwave Scanning Radiometer – Earth Observing System (AMSR-E), WindSat, Soil Moisture and Ocean Salinity (SMOS), and AMSR2 sensors. Active microwave products are from the Active Microwave Instrument Wind Scatterometer (AMI-WS), Advanced SCATterometer – satellite A or satellites A/B (ASCAT-A and ASCAT-A/B) sensors. Merging is performed by resampling target products to a uniform spatiotemporal structure, scaling to match ranges, and weighted using triple collocation. This research utilizes the COMBINED product. Resolutions are daily at 1/4° from 1978 to present for the globe* (Dorigo et al., 2017).

GPCP-v3          Global Precipitation Climatology Project, V3 Precipitation Data (Beta)

*Estimates of precipitation generated by assimilating radiometer data from SSMI and SSMIS, infrared data from the Precipitation Estimation from Remotely Sensed Information using Artificial Neural Networks – Climate Data Record (PERSIANN-CDR), TIROS Operational Vertical Sounder (TIROS-TOVS) sensor and Tropical Rainfall Measuring Missing (TRMM) Combined Climatology precipitation estimates with the GPCC global gauge analysis (see above in "Reanalysis"). This paper uses the "combined satellite-gauge" product rather than the "satellite-only" product. Resolutions are monthly at 1/2° from 1983 to 2016 for the globe* (Huffman et al., 2019).

MOD16-A2          MODIS Global Evapotranspiration Project

*Level 4 Moderate Resolution Imaging Spectroradiometer (MODIS) land data product that estimates evapotranspiration from ex-situ observational MODIS vegetation data and reanalysis meteorological data using an improved evapotranspiration model* (Mu et al., 2011) *based on the Penman-Monteith equation. Resolutions are 8-day at 1 km from 2000-2010 for the globe* (Running et al., 2017).

SMOS-L4          Soil Moisture and Ocean Salinity - Level 4



*Volumetric rootzone soil moisture estimates derived using CATDS-generated ascending and descending orbit SMOS L3 surface volumetric soil moisture estimates in conjunction with MODIS sensor-derived vegetation information, reanalysis NCEP climate data, FAO soil textures, and ECOCLIMAP surface cover. A double bucket model, composed of a simple water budget model (5-40 cm depth) and a budget model based on a linearized Richard's Equation formulation (40-200 cm depth), is used to extrapolate surface SMOS L3 soil moisture (0-5 cm) to the rootzone domain. This study merged the ascending and descending orbit products by mean. Resolutions are daily at 25 km EASE Grids from 2010 to 2017 for the globe (Al Bitar et al., 2013).*

SSEBop    Operational Simplified Surface Energy Balance model

*A parameterization of the Simplified Surface Energy Balance approach that estimates evapotranspiration from ex-situ MODIS vegetation, ex-situ Shuttle Radar Topography Mission (SRTM) elevation data, reanalysis meteorological data, and other modeled data using an energy balance approach where actual evapotranspiration is calculated as the difference between net surface radiation, sensible heat flux, and ground heat flux. Resolutions are monthly at 1 km from 2000 to 2014 for the conterminous United States and portions of surrounding countries (Senay et al., 2013, 2011).*

TMPA-3B43    TRMM Multisatellite Precipitation Analysis, V7

*From the larger Tropical Rainfall Measuring Mission (TRMM), generates spatiotemporally continuous precipitation estimates by merging precipitation estimates from numerous ex-situ data sources.  Primary data sources include (1) passive microwave data from various low earth orbit satellites (e.g. TRMM, AMSR-E) that are converted to precipitation estimates using source-specific algorithms; (2) infrared data collected by geosynchronous earth orbit satellites; (3) TRMM Combined Instrument estimate, a merged passive microwave and active radar product; 4) in-situ GPCP and CAMS monthly precipitation measurements. Resolutions are sub-daily at 1/4° from 1998 to present for the globe (Huffman et al., 2010, 2007).*





**Appendix 2**

Figures A2.1 – 2.10 (see below)



**Tables**

Table 1: Summary of datasets used in this research including assigned data category, abbreviated name, primary organization, literature reference, spatiotemporal resolution, and sourced hydrologic components. The reader is referred to Appendix 1 for definitions of abbreviated model names, as well as model descriptions and further references. Components are precipitation (P), actual evapotranspiration (AET), runoff (R), snow water equivalent (SWE), and rootzone soil moisture in equivalent water depth and volumetric water content (RZSME & RZSMV, respectively). Hydrologic models NMH-MWBM and -PRMS are based on a delineated (i.e. non-gridded), topographically derived spatial framework composed on Hydrologic Response Units (HRUs). The reanalysis product WaterWatch is generated at Hydrologic Units (HU) 2-8. The finest resolution product, HU8, is used in this study.

| Dataset | Group | Reference | Spatiotemporal | | Components |
|---|---|---|---|---|---|
| *Hydrologic Model* | | | | | |
| CPC | CPC | (Fan and van den Dool, 2004) | $1/2°$ | 1948-present | RZSME |
| CSIRO-PML[‡] | CSIRO | (Zhang et al., 2016) | $1/2°$ | 1981-2012 | AET |
| ERA5/H-TESSEL[‡] | ECMWF | (C3S, 2017) | $1/4°$ | 1979-present | R, SWE, RZSMV |
| ERA5-Land/H-TESSEL | ECMFW | (C3S, 2017) | $1/10°$ | 2001-present | AET, R, SWE, RZSMV |
| GLDAS-CLM[‡] | NASA | (Rodell et al., 2007a) | 1° | 1979-present | AET, R, SWE |
| GLEAM[*,‡] | U. of BE | (Martens et al., 2017) | $1/4°$ | 1980-present | AET, RZSMV |
| JRA-25/SiB | JMA | (Onogi et al., 2007) | 110km | 1979-present | R, SWE, RZSME/V |
| JRA-55/SiB[‡] | JMA | (Kobayashi et al., 2015) | 55km | 1957-present | R, SWE, RZSMV |
| Livneh-VIC | CIRES | (Livneh et al., 2013) | $1/16°$ | 1915-2011 | SWE |
| MERRA-Land/CLSM[‡] | NASA | (Reichle et al., 2011) | $1/2°$ | 1980-2016 | AET, R, SWE |
| MERRA-2/CLSM[‡] | NASA | (Reichle et al., 2017) | $1/2°$ | 1980-present | AET, R, SWE, RZSMV |
| NCEP-DOE/Eta-Noah[‡] | NOAA | (Kanamitsu et al., 2002) | 210 km | 1979-present | R, SWE |
| NCEP-NARR/Eta-Noah | NOAA | (Mesinger et al., 2006) | 32 km | 1979-present | SWE |
| NHM-MWBM[‡] | USGS | (McCabe and Markstrom, 2007) | HRU | 1949-2010 | AET, R, SWE, RZSME |
| NHM-PRMS[‡] | USGS | (Regan et al., 2018) | HRU | 1980-2016 | AET, R, SWE, RZSME |
| NLDAS2-Mosaic[‡] | NASA | (Y. Xia et al., 2012b) | $1/8°$ | 1979-present | AET, R, SWE, RZSME |
| NLDAS2-Noah[‡] | NASA | (Y. Xia et al., 2012a) | $1/8°$ | 1979-present | AET, R, SWE, RZSME |
| NLDAS2-VIC[‡] | NASA | (Y. Xia et al., 2012c) | $1/8°$ | 1979-present | AET, R, SWE, RZSME |
| SNODAS | NWS | (Barrett, 2003) | 1km | 2003-present | SWE |
| TerraClimate[‡] | U. of ID | (Abatzoglou et al., 2018) | $1/24°$ | 1958-present | AET, R, SWE, RZSME |
| VegET[‡] | USGS | (Senay, 2008) | 1 km | 2000-2014 | AET, RZSME |
| *Reanalysis* | | | | | |
| CanSISE | U. Toronto | (Mudryk and Derksen, 2017) | 1° | 1981-2010 | SWE |
| CMAP[†,‡] | CPC | (Xie and Arkin, 1997) | 2 ½° | 1979-present | P |
| DayMET[‡] | ORNL | (Thornton et al., 2018) | 1 km | 1980-present | P, SWE |
| ERA5[‡] | ECMWF | (C3S, 2017) | $1/4°$ | 1979-present | P |
| ERA5-Land | ECMWF | (C3S, 2019) | $1/10°$ | 2001-present | P |
| GPCC[‡] | GPCC | (Becker et al., 2013) | $1/2°$ | 1901-2013 | P |



| | | | | | | |
|---|---|---|---|---|---|---|
| gridMET‡ | U. of ID | (Abatzoglou, 2013) | $1/24°$ | 1979-present | P |
| Livneh et al. 2013‡ | NOAA | (Livneh et al., 2013) | $1/16°$ | 1915-2011 | P |
| Maurer et al. 2002 | U. WA | (Maurer et al., 2002) | $1/8°$ | 1950-1999 | P |
| MERRA-Land‡ | NASA | (Rienecker et al., 2011) | $1/2°$ | 1980-2016 | P |
| MERRA-2‡ | NASA | (Gelaro et al., 2017) | $1/2°$ | 1980-present | P |
| NCEP-DOE‡ | NCEP/DOE | (Kanamitsu et al., 2002) | 210 km | 1979-present | P |
| NLDAS2‡ | NASA | (Xia, NCEP/EMC, et al., 2009) | $1/8°$ | 1979-present | P |
| PRISM‡ | OSU | (PRISM Climate Group, 2004) | 4 km | 1895-present | P |
| Reitz et al. 2017‡ | USGS | (Reitz et al., 2017) | 800 m | 2000-2013 | AET |
| UoD-v5‡ | U. of DE | (Willmott and Matsuura, 2001) | $1/2°$ | 1950-1999 | P |
| WaterWatch‡ | USGS | (Jian et al., 2008) | HU8 | 1901-present | R |
| *Remote Sensing* | | | | | |
| AMSR-E/Aqua | NSIDC | (Tedesco et al., 2004) | 25 km | 2002-2011 | SWE |
| ESA-CCI | ESA | (Dorigo et al., 2017) | $1/4°$ | 1978-present | RZSMV |
| GPCP-v3‡ | NASA | (Huffman et al., 2019) | $1/2°$ | 1983-2016 | P |
| MOD16-A2‡ | U. of MT | (Running et al., 2017) | 1 km | 2000-2010 | AET |
| SMOS-L4 | FNCSR | (Al Bitar et al., 2013) | 25 km | 2010-2017 | RZSMV |
| SSEBop‡ | USGS | (Senay et al., 2013) | 1 km | 2000-2014 | AET |
| TMPA-3B43‡ | NASA | (Huffman et al., 2010) | $1/4°$ | 1998-present | P |

*\* Versions 3.3a, & 3.3b*

*† Versions Standard & Enhanced*

*‡ Datasets used in the water budget case study with complete data for water years 2001-2010*

**Table 1: Sizes of study ecoregions in square kilometres [km²] and percent of study domain.**

| Ecoregion | Area [km²] | Percent of CONUS [%] |
|---|---|---|
| Marine West Coast Forest | 91,035 | 1.11 |
| Mediterranean California | 168,757 | 2.06 |
| Northwestern Forested Mountains* | 859,801 | 10.52 |
| North American Deserts* | 1,533,536 | 18.76 |
| Southern Semiarid Highlands | 66,699 | 0.82 |
| Temperate Sierras | 110,038 | 1.35 |
| Great Plains* | 2,328,673 | 28.48 |
| Eastern Temperate Forests* | 2,561,292 | 31.33 |
| Northern Forests | 433,032 | 5.30 |
| Tropical Wet Forests | 22,469 | 0.27 |
| **Total Domain Area** | **8,175,332** | **100%** |

*\*Primary ecoregions highlighted during water budget analysis*




**Table 3: Spearman's rho (ρ) correlation of monthly hydrologic model and reanalysis dataset estimates against remote sensing products quantifying hydrologic components precipitation (P), actual evapotranspiration (AET), snow water equivalent (SWE), and rootzone soil moisture (RZSM). Correlation was calculated at ten ecoregions and the CONUS extent. The number of modelled datasets correlated against remote sensing products is shown as (n = X) for each column. Mean and standard deviation (SD) of ρ are provided for each ecoregion and remote sensing products as ρ ± SD. Counts of statistically insignificant correlations (Y) are denoted when n > 0 as (Y). RZSM combines datasets produced in units of equivalent water depth (RZSME in text) and volumetric soil moisture water content (RZSMV in text). Correlation of SWE products is included only for regions of high annual mean SWE storage. Hydrologic model and reanalysis datasets (Table 1) used in each column are listed in the footnotes.**


| | P | | AET | | SWE | RZSM | |
|---|---|---|---|---|---|---|---|
| Ecoregion | GPCP-v3[1] (n=15) | TMPA-3B43[2] (n=14) | MOD16-A2[3] (n=15) | SSEBop[3] (n=15) | AMSR-E/Aqua[4] (n=18) | ESA-CCI[5] (n=15) | SMOS-L4[6] (n=13) |
| Marine W. Coast Forest | 0.99 ± 0.01 | 0.99 ± 0.01 | 0.82 ± 0.18 | 0.80 ± 0.19 | * | 0.72 ± 0.09 | 0.78 ± 0.09 |
| Medit. California | 0.98 ± 0.01 | 0.97 ± 0.02 | 0.71 ± 0.20 | 0.25 ± 0.42 *(4)* | * | 0.79 ± 0.13 | 0.92 ± 0.04 |
| NW Forested Mtns | 0.97 ± 0.05 | 0.96 ± 0.05 | 0.91 ± 0.06 | 0.89 ± 0.08 | 0.91 ± 0.07 | 0.58 ± 0.10 | 0.27 ± 0.12 *(2)* |
| N. American Deserts | 0.95 ± 0.05 | 0.94 ± 0.05 | -0.28 ± 0.17 *(4)* | 0.81 ± 0.13 | * | 0.61 ± 0.18 | 0.28 ± 0.11 *(2)* |
| So. Semiarid Highlands | 0.97 ± 0.04 | 0.96 ± 0.05 | 0.67 ± 0.17 | 0.64 ± 0.17 | * | 0.63 ± 0.20 | 0.79 ± 0.16 |
| Temperate Sierras | 0.97 ± 0.04 | 0.96 ± 0.04 | 0.59 ± 0.15 | 0.68 ± 0.18 | * | 0.51 ± 0.19 | 0.43 ± 0.21 *(1)* |
| Great Plains | 0.99 ± 0.02 | 0.98 ± 0.01 | 0.92 ± 0.02 | 0.93 ± 0.03 | * | 0.62 ± 0.15 | 0.69 ± 0.06 |
| E. Temperate Forests | 0.95 ± 0.09 | 0.94 ± 0.07 | 0.96 ± 0.03 | 0.94 ± 0.03 | * | 0.74 ± 0.10 | 0.83 ± 0.08 |
| Northern Forests | 0.96 ± 0.03 | 0.93 ± 0.02 | 0.94 ± 0.03 | 0.94 ± 0.02 | 0.80 ± 0.03 | 0.44 ± 0.08 | 0.12 ± 0.21 *(10)* |
| Tropical Wet Forests | 0.97 ± 0.03 | 0.96 ± 0.03 | 0.88 ± 0.05 | 0.88 ± 0.09 | * | 0.40 ± 0.19 *(1)* | 0.57 ± 0.16 |
| *CONUS‡* | 0.94 ± 0.07 | 0.91 ± 0.06 | 0.95 ± 0.02 | 0.95 ± 0.03 | 0.91 ± 0.03 | 0.17 ± 0.05 *(4)* | 0.68 ± 0.09 |

[1] *CMAP-Enhanced & -Standard, DayMET, ERA5, ERA5-Land, GPCC, gridMET, Livneh et al. 2013, Maurer et al. 2002, MERRA-2, MERRA-Land, NCEP-DOE, NLDAS2, PRISM, and UoD-v5.*

[2] *CMAP-Enhanced & -Standard, DayMET, ERA5, ERA5-Land, GPCC, gridMET, Livneh et al. 2013, MERRA-2, MERRA-Land, NCEP-DOE, NLDAS2, PRISM, and UoD-v5.*

[3] *CSIRO-PML, ERA5-Land/H-TESSEL, GLDAS-CLM, GLEAM-v3.3a & -v3.3b, MERRA-2 & MERRA-Land/CLSM, NHM-MWBM & -PRMS, NLDAS2-Mosaic, -Noah, & -VIC, Reitz et al. 2017, TerraClimate, and VegET.*

[4] *CanSISE, DayMET, ERA5-Land/H-TESSEL, GLDAS-CLM, JRA-25 & JRA-55/SiB, Livneh-VIC, MERRA-2 & MERRA-Land/CLSM, NCEP-DOE & NCEP-NARR/Eta-Noah, NHM-MWBM & -PRMS, NLDAS2-Mosaic, -Noah, & -VIC, SNODAS, and TerraClimate.*

[5] *CPC, ERA5-Land & ERA5/H-TESSEL, GLEAM-v3.3a & -v3.3b, JRA-25 & JRA-55/SiB, MERRA-2/CLSM, NHM-MWBM & -PRMS, NLDAS2-Mosaic, -Noah & -VIC, TerraClimate, and VegET.*

[6] *CPC, ERA5-Land & ERA5/H-TESSEL, GLEAM-v3.3a & -v3.3b, JRA-55/SiB, MERRA-2/CLSM, NHM-PRMS, NLDAS2-Mosaic, -Noah, & -VIC, TerraClimate, and VegET.*

\* *Regions omitted.*

‡ *Calculated from datasets spatially aggregated by weighted area mean at the CONUS extent (Figure 1), not as a mean of individual ecoregion ρ estimates above.*





**Table 4: Summary statistics of ecoregion water budget calculations summed for complete water years 2001-2010. Median, mean, and standard deviation (SD) summarize the iterative water budgets' (n = 2,925) relative residuals (Rε) in units of percent. The 10th, 25th, 75th, and 90th percentiles (Pi) quantify distribution ranges. The ensemble column provides water budget Rε produced from the ensemble mean of all datasets available for water years 2001-2010.**

| Ecoregion | SD | $P_{10}$ | $P_{25}$ | Median | $P_{75}$ | $P_{90}$ | Ensemble | Skew | Kurtosis |
|---|---|---|---|---|---|---|---|---|---|
| Marine W. Coast Forest | ±29.5 | -46.0 | -21.3 | -2.3 | 12.2 | 26.8 | -3.8 | -0.9 | 1.1 |
| Medit. California | ±27.0 | -45.4 | -18.6 | -3.6 | 9.8 | 23.0 | -4.7 | -1.0 | 1.3 |
| **NW Forested Mtns*** | **±45.6** | **-83.3** | **-35.0** | **-6.7** | **10.5** | **23.8** | **-13.0** | **-1.7** | **4.3** |
| **N. American Deserts*** | **±27.0** | **-33.2** | **-13.7** | **4.7** | **21.1** | **36.6** | **3.8** | **-0.5** | **0.3** |
| So. Semiarid Highlands | ±30.0 | -40.5 | -7.2 | 4.9 | 19.0 | 45.2 | 5.8 | -0.7 | 1.5 |
| Temperate Sierras | ±37.2 | -64.6 | -15.5 | 3.0 | 14.9 | 31.4 | 0.4 | -1.4 | 2.2 |
| **Great Plains*** | **±18.2** | **-23.2** | **-9.1** | **3.9** | **14.8** | **23.6** | **2.5** | **-0.6** | **0.5** |
| **E. Temperate Forests*** | **±14.3** | **-19.9** | **-10.4** | **-0.4** | **8.8** | **17.2** | **-0.5** | **-0.2** | **-0.1** |
| Northern Forests | ±18.9 | -24.3 | -13.0 | -0.2 | 12.6 | 23.9 | 0.4 | -0.1 | 0.0 |

*Primary focus ecoregions


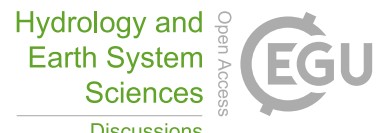

## Figures

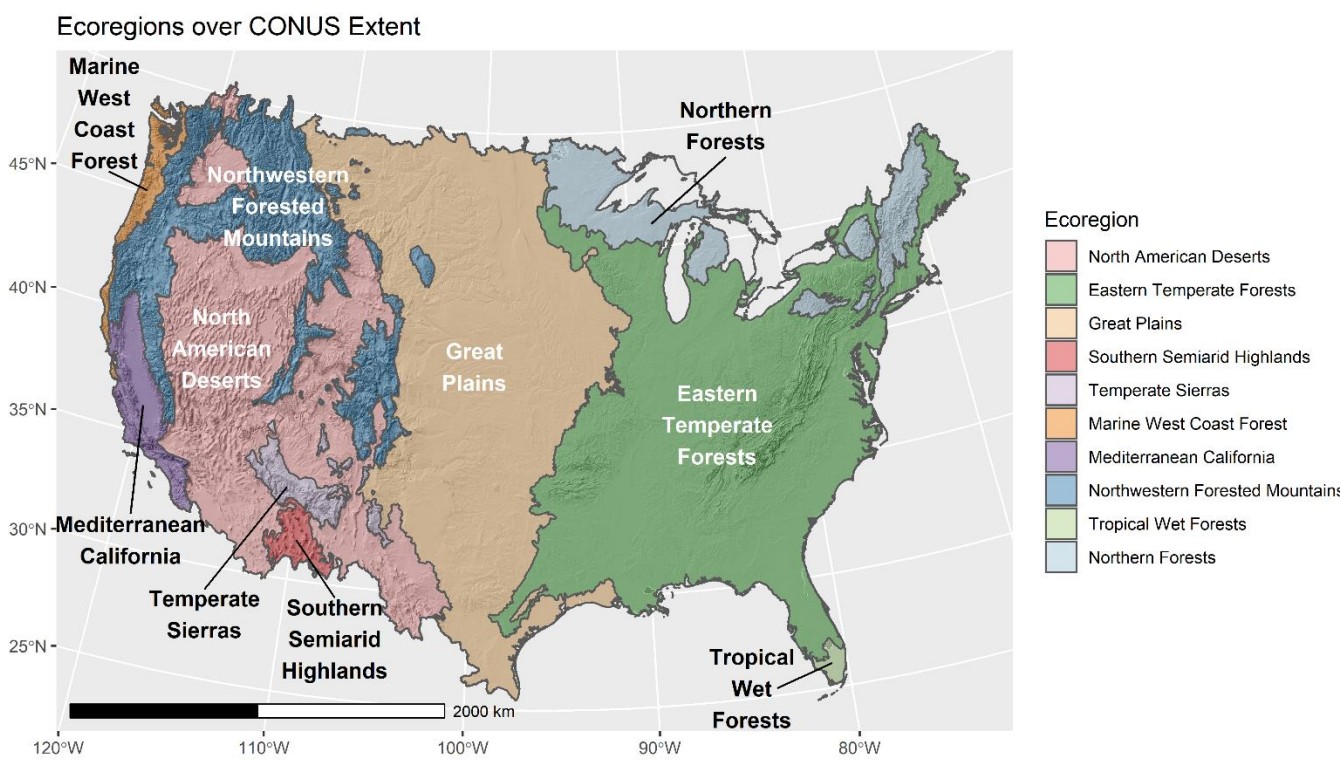

**Figure 1: Distribution of the ten ecoregions covering the CONUS study domain.**





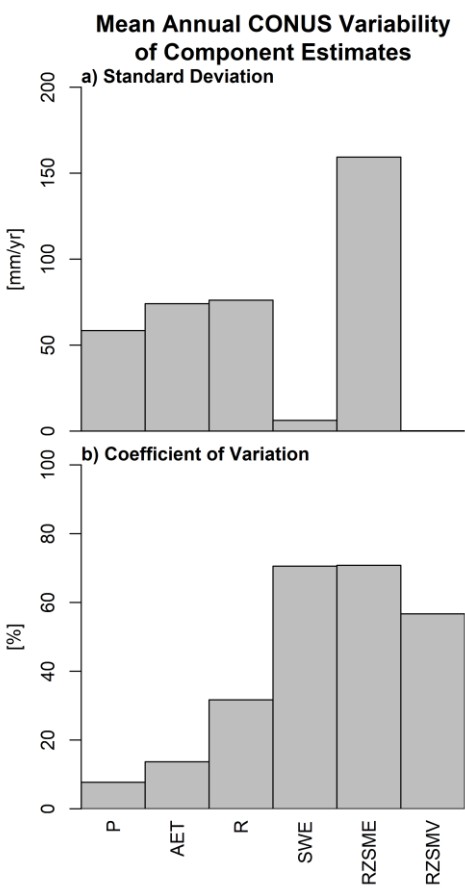


**Figure 2: Mean inter-annual a) standard deviation and b) coefficient of variation calculated between CONUS-extent modeled estimates of hydrologic components precipitation (P), actual evapotranspiration (AET), runoff (R), snow water equivalent (SWE), rootzone soil moisture in equivalent water depth (RZSME), and rootzone soil moisture in volumetric water content (RZSMV).**



**Figure 3: Boxplots of annual water year magnitudes of all study model estimates averaged over the CONUS extent for hydrologic components precipitation (P), actual evapotranspiration (AET), runoff (R), snow water equivalent (SWE), rootzone soil moisture in units of equivalent water depth (RZSME), and rootzone soil moisture in units of volumetric water content (RZSMV). Dataset magnitudes are subdivided into two periods, 1985-1999 (left) and 2000-2014 (right). Flux component estimates (P, AET, R) are summed from monthly values to calculate annual water year rates. Storage component estimates (SWE, RZSME, RZSMV) are averaged from monthly values to calculate annual water year average storage values. Box lower limits, midlines, and upper limits represent the 25th, 50th (median), and 75th percentiles, respectively, of the associated data. Whiskers represent 1.5 times the interquartile range. Box color denotes dataset categories of hydrologic model, reanalysis, or remote sensing-derived.**





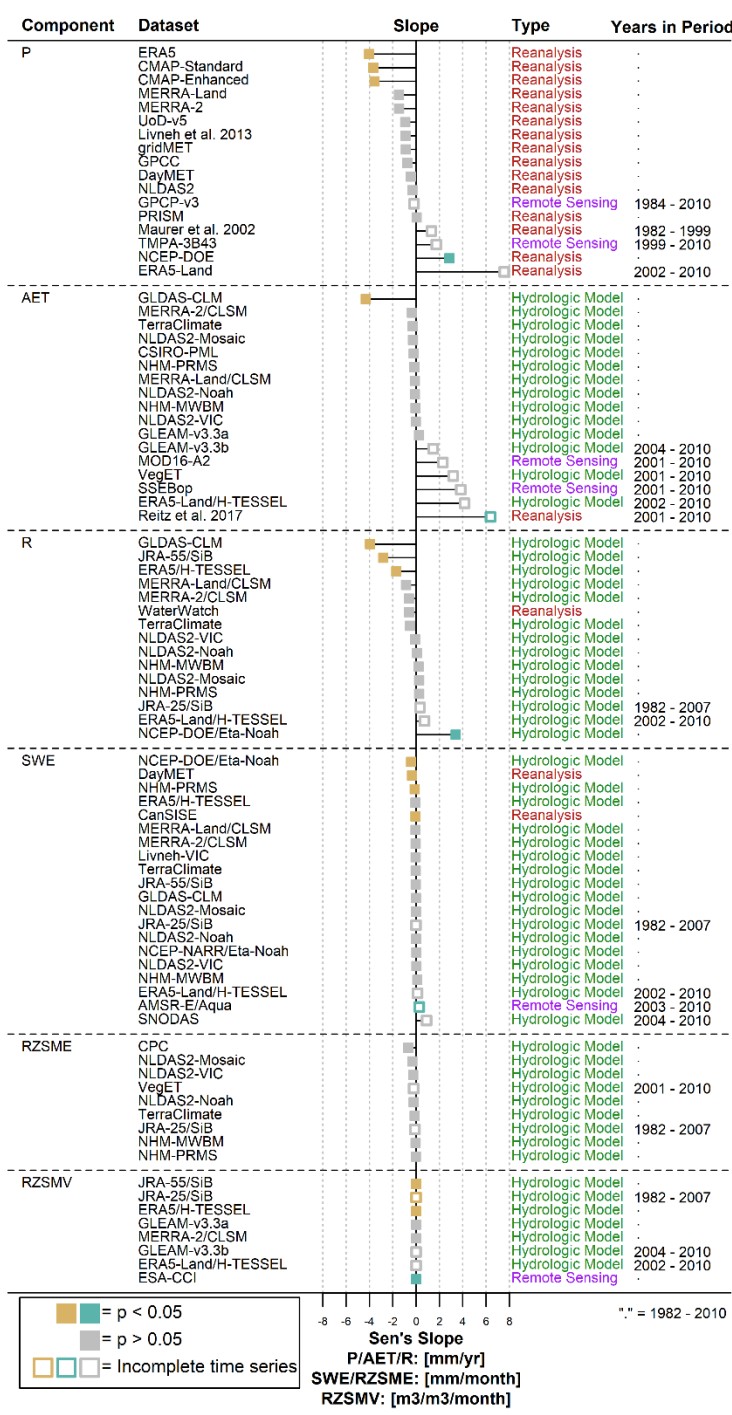


**Figure 4: Forest plot of inter-annual (water year) component estimate Sen's slope from 1982-2010. Insignificant trends (p > 0.05) are grey. Significant trends are colored based on direction: negative trends are gold, positive trends are green. Datasets without**





complete data during the study period are represented with hollow points and temporal extent is noted under "Years in Period" column.


**Figure 5: Boxplots of annual coefficient of variation (CV) between component estimates at ten ecoregions. CV distributions are subdivided into two periods, 1985-1999 and 2000-2014. Box lower limits, midlines, and upper limits represent the 25th, 50th (median), and 75th percentiles, respectively, of the associated data. Whiskers represent 1.5 times the interquartile range. Box colors denote ecoregion and correspond to map colors in Figure 1.**


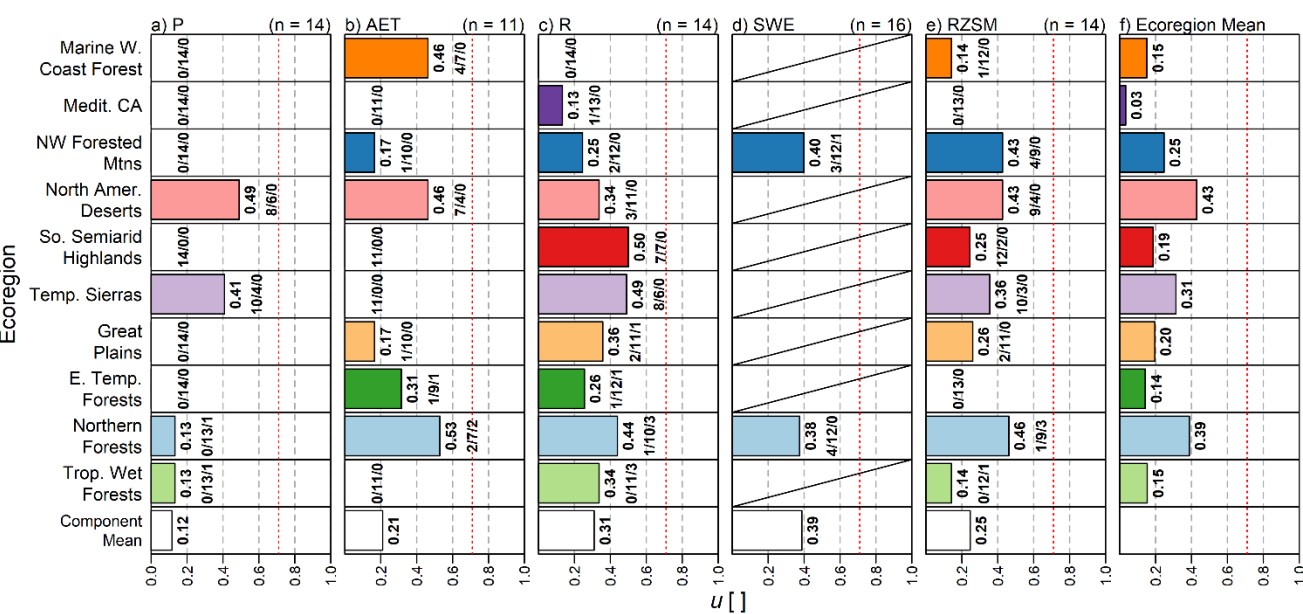

**Figure 6: Bar plots summarizing the disagreement in Mann-Kendall trend ($\tau$) direction using the unalikeability coefficient ($u$) after categorizing $\tau$ as significant negative trend, significant positive trend, or no significant trend for ten ecoregions from 1982-2010. Trends were assumed to be significant is $p < 0.05$. $u$ can range from complete agreement ($u = 0$) to complete disagreement ($u = 1$) when sample size ($n$) is less than the number of categorical values ($c$). Because here $n > c$, maximum possible $u$ is approximately 0.71 (dotted red line). Text to the right of each bar shows, in order, the $u$ value and the number of datasets showing negative/insignificant/positive trend. Bars are color-matched to ecoregion colors from Figure 1 and are ordered from west (top) to east (bottom).**

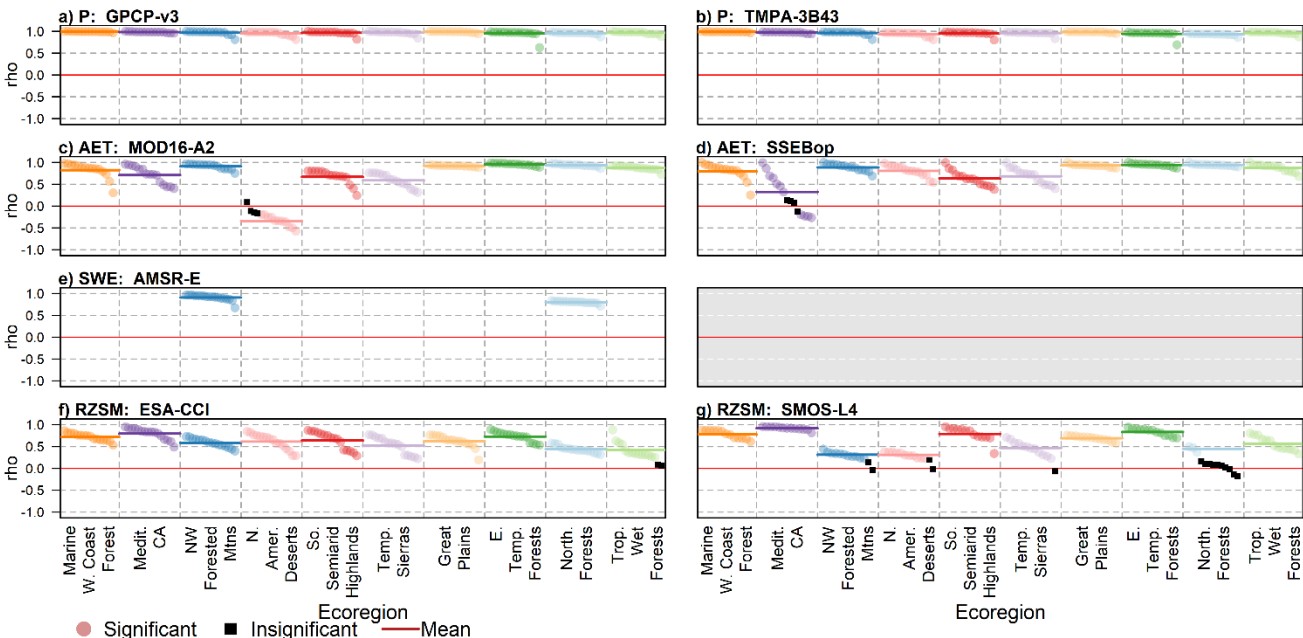

**Figure 7: Spearman's rho (ρ) correlation values of hydrologic model and reanalysis dataset component estimates against remote sensing-derived products. The title of each sub-plot (e.g. Figure 7a) provides the hydrologic component (e.g. precipitation) and the remote sensing product against which the dataset was correlated (e.g. GPCP-v3). Statistically significant ρ values (p < 0.05) are shown as colored circles. Insignificant ρ values (p > 0.05) are shown as black squares. Horizontal bars within each ecoregion denote mean ρ. Point and bar colors correspond to ecoregions, as shown in Figure 1.**





## Histograms of Iterative Relative
## Water Budget Residuals by Ecoregion,
## WY 2001-2010

**Figure 8: Histograms of eight ecoregion water budget relative residuals calculated from combinations of all annual P (15), AET**

**(15), and R (13) datasets with complete data from water years 2001-2010 (Table 1). Each region yielded 2,925 water budgets. The 10th and 90th percentiles of each ecoregion distribution are provided as fine vertical dotted lines. Boxplots are overlain, showing the median (black bar), 25th and 75th percentiles (box hinges), interquartile range (IQR) (box), and 1.5xIQR (whiskers). Bold and coarse dashed vertical lines represent the relative residuals of 10-year (2001-2010) water budgets from the ensemble mean of all modeled products.**






**Figure A2.1: Boxplots of annual standard deviation (SD) between component estimates at ten ecoregions. SD distributions are subdivided into two periods, 1985-1999 and 2000-2014. Box lower limits, midlines, and upper limits represent the 25th, 50th (median), and 75th percentiles, respectively, of the associated data. Whiskers represent 1.5 times the interquartile range. Box colors denote ecoregions and correspond to map colors in Figure 1.**






**Figure A2.2: Boxplot summarizing annual peak snow water equivalent (SWE) depths by ecoregion derived from the distribution of modeled estimates used in this papers. Box lower limits, midlines, and upper limits represent the 25th, 50th (median), and 75th percentiles, respectively, of the associated data. Whiskers represent 1.5 times the interquartile range.**




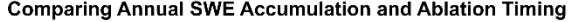

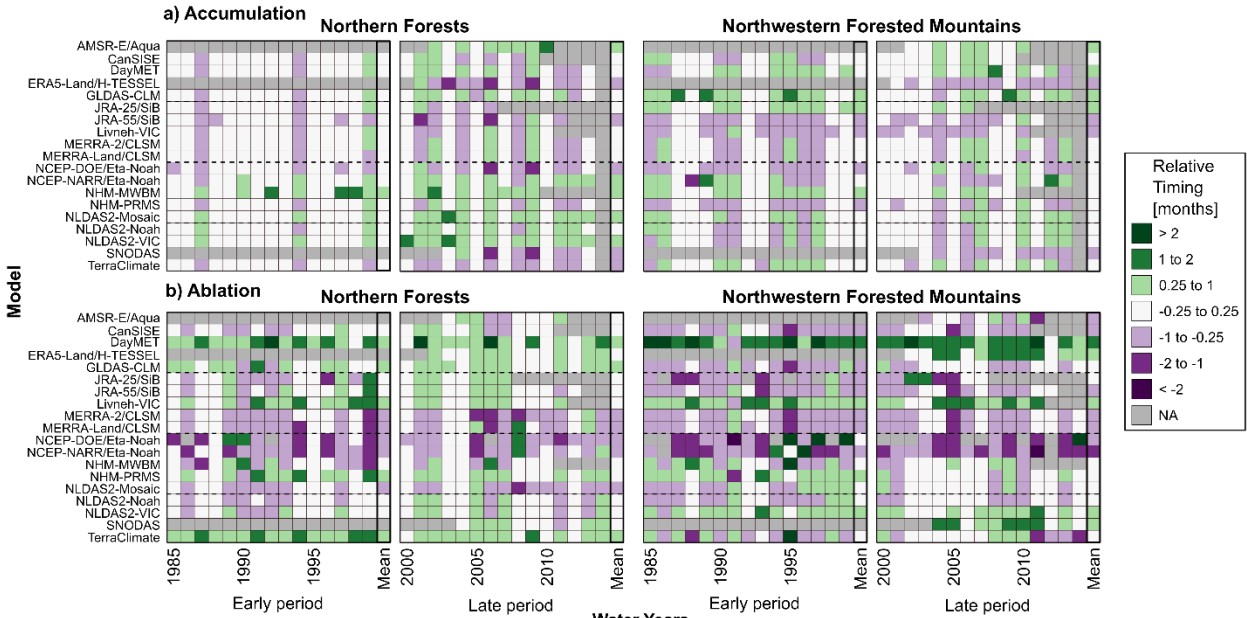

**Figure A2.3:** Heatmaps showing the relative timing of the beginning of snow water equivalent (SWE) accumulation (A2.3a) and ablation (A2.3b) periods for two ecoregions from 1985-2014. Relative timing is the difference between the month of the beginning of a model's accumulation or ablation period and the mean of all other model's timing. The beginning of accumulation and ablation is defined as when the rate of change of SWE between months is greater than 1 mm/month or less than -1 mm/month, respectively.






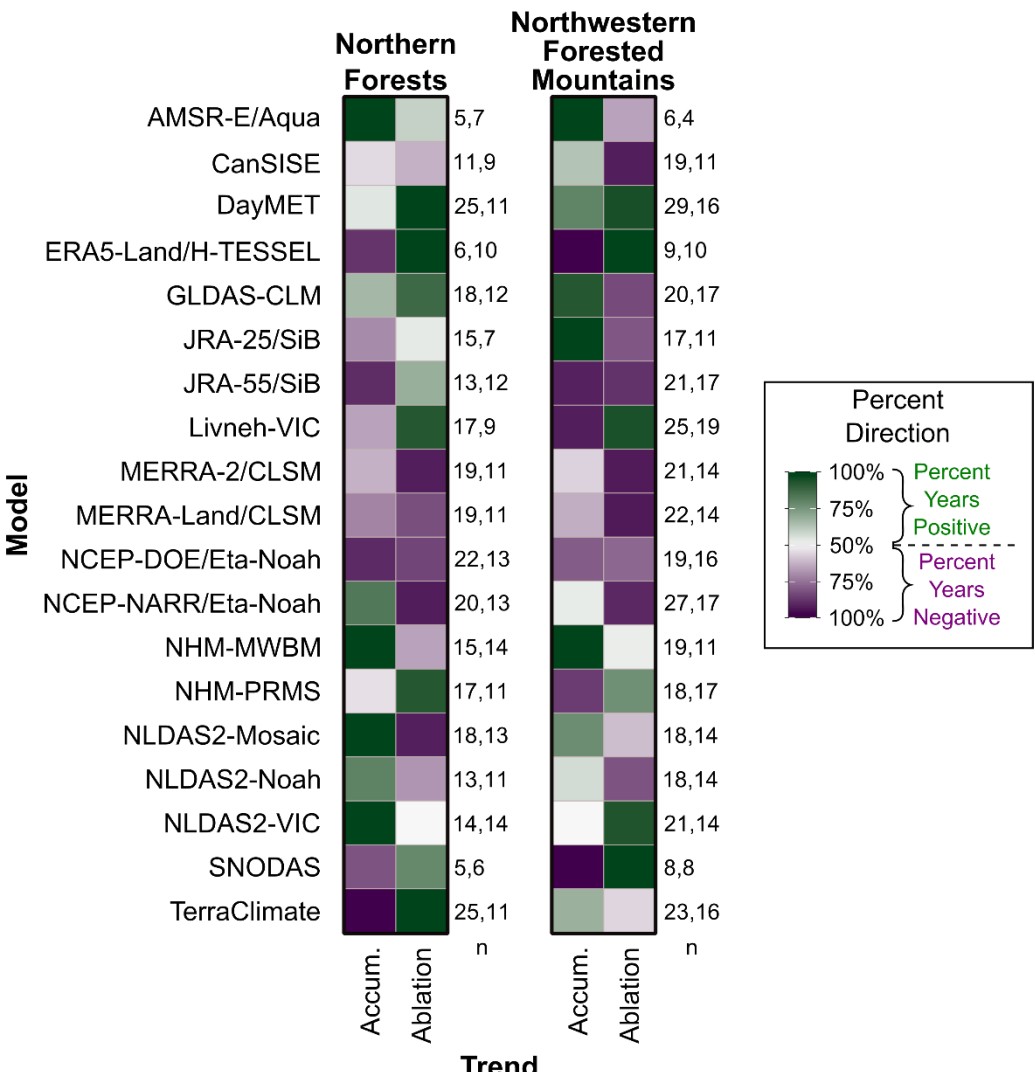

**Figure A2.4: Heatmaps summarizing the annual relative timing values of SWE accumulation (accum.) and ablation from A2.3 for two ecoregions. Grid cells are colored by the percentage of years that have a positive relative timing (green) or by the percentage of years that have a negative relative timing (purple), whichever is greater.**



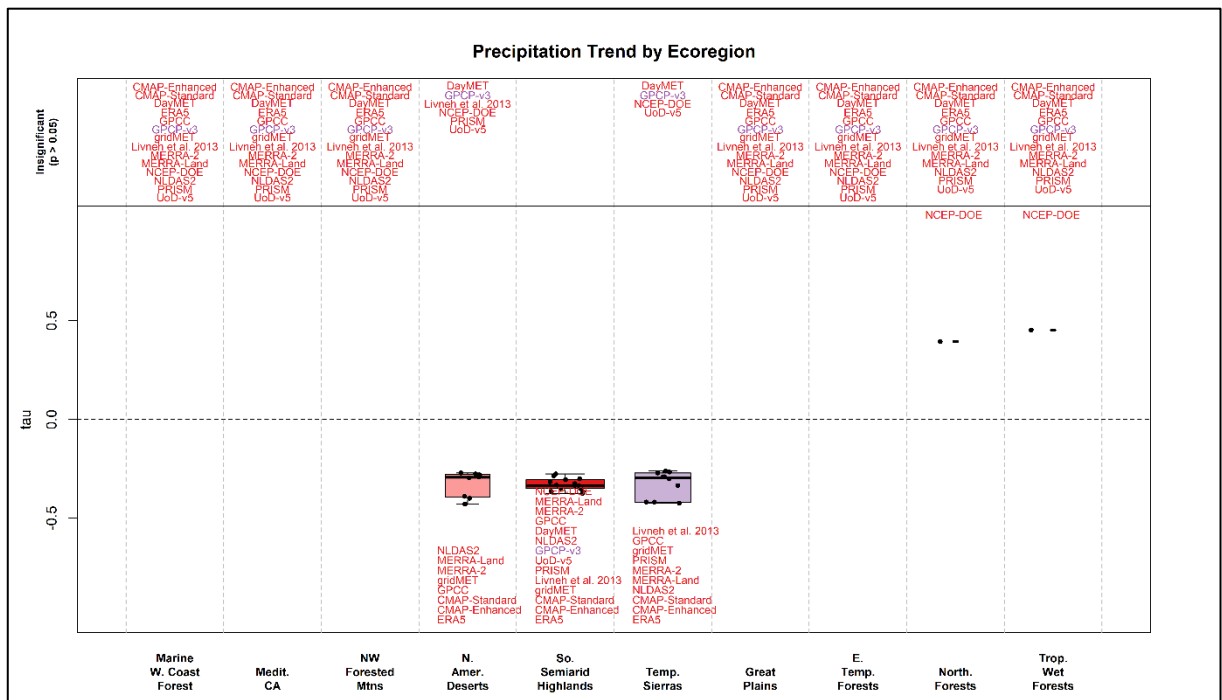

**Figure A2.5: Boxplots of precipitation estimate Mann-Kendall tau (τ) values from 1982-2010 for 10 ecoregions. Box lower limits, midlines, and upper limits represent the 25th, 50th (median), and 75th percentiles, respectively, of the associated data. Whiskers represent 1.5 times the interquartile range. Datasets with insignificant trends (p > 0.05) are listed on top. Datasets with significant trends are included within each ecoregion, ordered by magnitude of trend. Text color denotes product category: green – hydrologic model, red – reanalysis, and purple – remote sensing.**




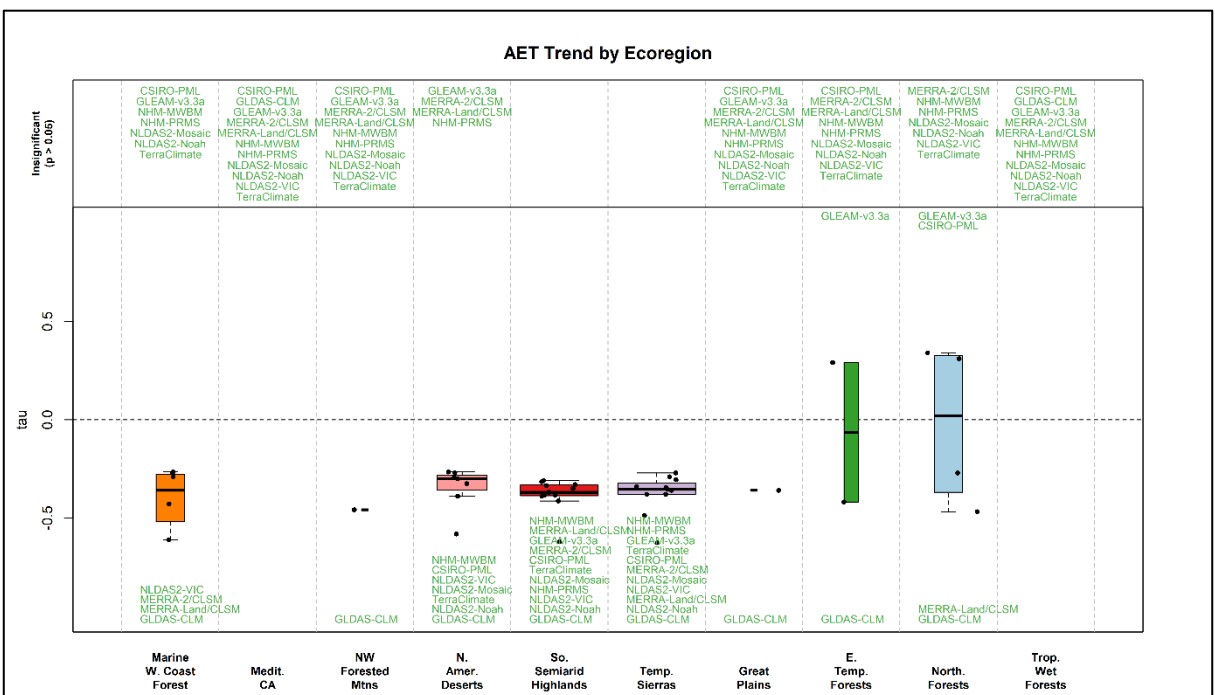


**Figure A2.6: Boxplots of evapotranspiration estimate Mann-Kendall tau (τ) values from 1982-2010 for 10 ecoregions. Box lower limits, midlines, and upper limits represent the 25th, 50th (median), and 75th percentiles, respectively, of the associated data. Whiskers represent 1.5 times the interquartile range. Datasets with insignificant trends (p > 0.05) are listed on top. Datasets with significant trends are included within each ecoregion, ordered by magnitude of trend. Text color denotes product category: green – hydrologic model, red – reanalysis, and purple – remote sensing.**




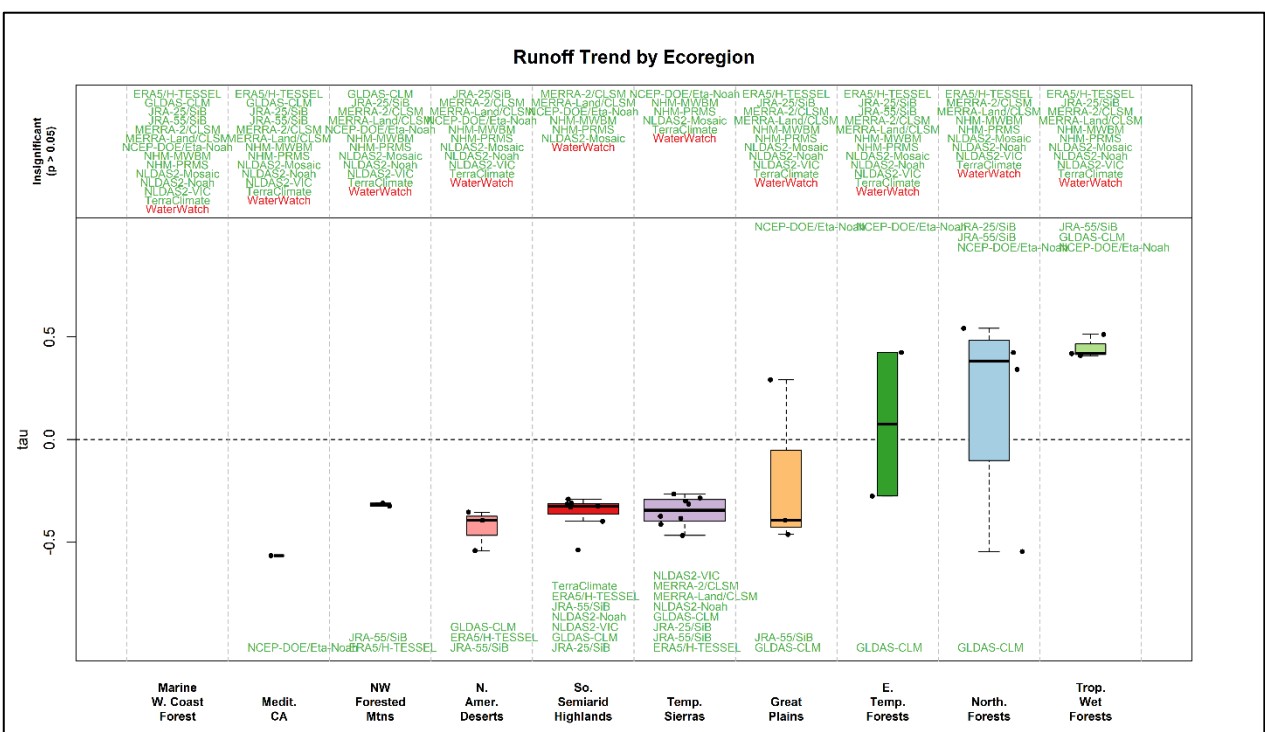

**Figure A2.7: Boxplots of runoff estimate Mann-Kendall tau (τ) values from 1982-2010 for 10 ecoregions. Box lower limits, midlines, and upper limits represent the 25th, 50th (median), and 75th percentiles, respectively, of the associated data. Whiskers represent 1.5 times the interquartile range. Datasets with insignificant trends (p > 0.05) are listed on top. Datasets with significant trends are included within each ecoregion, ordered by magnitude of trend. Text color denotes product category: green – hydrologic model, red – reanalysis, and purple – remote sensing.**





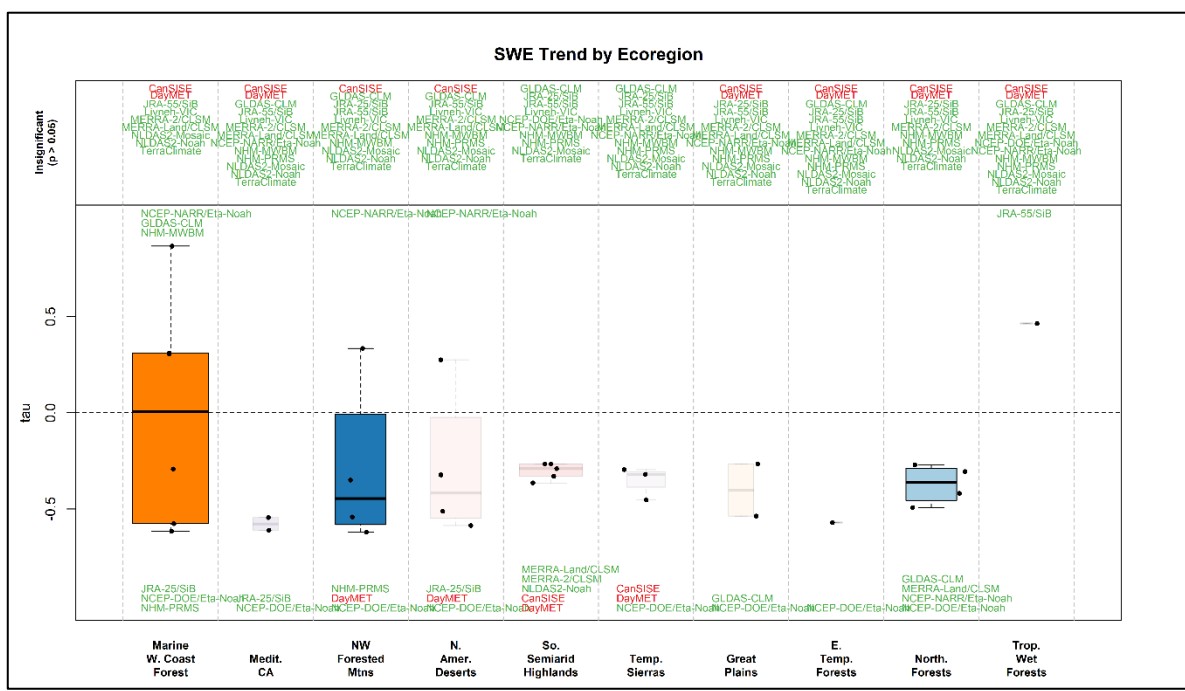

**Figure A2.8: Boxplots of snow water equivalent (SWE) estimate Mann-Kendall tau (τ) values from 1982-2010 for 10 ecoregions. Box lower limits, midlines, and upper limits represent the 25th, 50th (median), and 75th percentiles, respectively, of the associated data. Whiskers represent 1.5 times the interquartile range. Datasets with insignificant trends (p > 0.05) are listed on top. Datasets with significant trends are included within each ecoregion, ordered by magnitude of trend. Text color denotes product category: green – hydrologic model, red – reanalysis, and purple – remote sensing.**

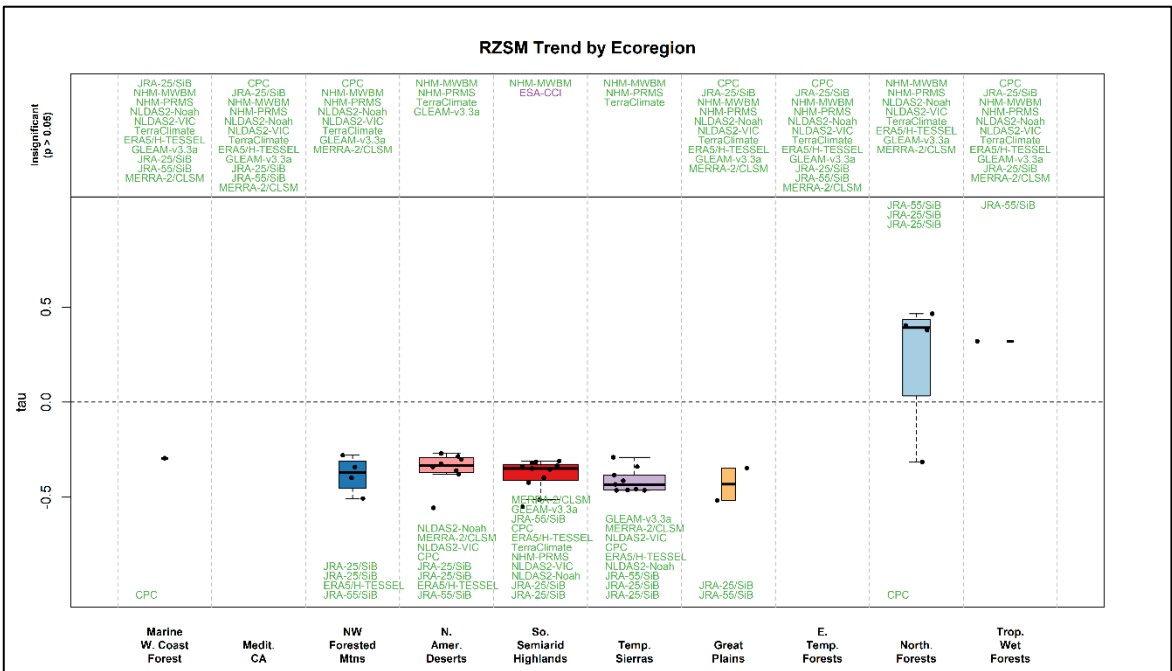

**Figure A2.9:** Boxplots of rootzone soil moisture estimate Mann-Kendall tau (τ) values from 1982-2010 for 10 ecoregions. Box lower limits, midlines, and upper limits represent the 25th, 50th (median), and 75th percentiles, respectively, of the associated data. Whiskers represent 1.5 times the interquartile range. Datasets with insignificant trends (p > 0.05) are listed on top. Datasets with significant trends are included within each ecoregion, ordered by magnitude of trend. Text color denotes product category: green – hydrologic model, red – reanalysis, and purple – remote sensing.





**Figure A2.10: Distribution of hydrologic model and reanalysis dataset correlation against remote sensing products using Spearman's rho (ρ), provided by water budget component. Values of ρ are binned by > 0.90, 0.50-0.90, <0.50, and statistically**



insignificant (p > 0.05). Rectangles are color-paired with their associated ecoregions (Fig. 1). Water budget components shown are precipitation (precip), actual evapotranspiration (AET), snow water equivalent (SWE), and rootzone soil moisture (RZSM).