# Peer review of "Implications of Model Selection: A Comparison of Publicly Available, CONUS-Extent Hydrologic Component Estimates"

_Hydrology and Earth System Sciences, 2020_

## Referee Comment (RC1) · Janneke Remmers (Referee) · 2 Jul 2020

Summary

This study evaluates the effects of model disagreement for publicly available estimates of the hydrological cycle. These estimates were created by hydrologic modelling, re-analysis and remote sensing. Multiple statistical methods were employed to perform this comparison, e.g. the coefficient of variation, the Mann-Kendall trend test, Sen's slope and the Spearman's Rho test. The datasets were compared based on precipitation, actual evapotranspiration, root zone soil moisture and discharge. The study focuses strongly on the Land Surface Models in the results, discussion and conclu-

sion. The main conclusions are that the disagreement between models varies spatially and the authors recommend the use of model ensembles. In my opinion, this study suits HESS well.

Major comments:

I think this study supplements the previous studies concerning the evaluation of publicly available datasets. The use of literature is extensive, which supports why this study supplements the existing science. In the introduction this is done thoroughly and in detail. I have understood that there are two reasons: discharge has not been compared that often and the comparison is often made with in-situ observations, not between the models. In the conclusion, the reasons why this study is supplementary is revisited, though mainly the latter reason. Within the results and discussion, I missed this focus. In both sections, it seems to me that the focus is put on P and AET. While R has received less attention in the past, it was not discussed as much as I expected it to be. For example, Fig. 7 does not contain a subplot with R. It seems that a subplot can be easily added, since one subplot is empty.

The study has been set up well, in generally. The use of different statistical methods complement each other well. There are three aspects that can either be improved or be stated more clearly.

1) I think the readability would be improved if a more clear structure would be implemented in the statistical part of the methods, as is already done in the results. I would suggest to use subsections and state more clearly how each subsection contributes to the main goal and relates to the other subsections.

2) I was confused by the different time periods applied for the analysis. By implementing a clearer structure, I think this will be partly remedied. On top of that, I would suggest to clarify the purpose of each time period more clearly.

3) The relative imbalance is calculated for 8 out of 10 ecoregions in the cases studies.

Two regions, "Tropical west forests" and "Southern semi-arid highlands", are excluded due to their relatively limited extent, respectively 0.27% and 0.82% of the total CONUS land surface. Yet, three other ecoregions ("Temperate sierras", "Marine west coast forest" and "Mediterranean California") have a relatively limited extent too (respectively 1.11%, 1.35% and 2.06%). Thus, I wondered why the two former ecoregions are excluded, but the latter three are not. Concerning the imbalances, the terminology of 'imbalances' is proper, as opposed to using 'residuals'. The implicit assumption here is that the residuals, i.e. model uncertainty, are considered constant for every case study. Clark et al. (2008) show that model uncertainty differs between climates. Especially, more arid climate have higher residuals. Thus, the analysis based on the relative imbalance can only be semi-reasonably compared within 1 ecoregion. I do think imbalances is an interesting concept to use.

Clark, M. P., Slater, A. G., Rupp, D. E., Woods, R. A., Vrugt, J. A., Gupta, H. V., ... & Hay, L. E. (2008). Framework for Understanding Structural Errors (FUSE): A modular framework to diagnose differences between hydrological models. Water Resources Research, 44(12).

As mentioned above, the originality of the study can be shown more clearly in the results/discussion. Aside from that, I would suggest to critically examine both sections as well regarding the text and the figures. Some aspects of the analysis seem odd to me. For example, line 464 states "Runoff datasets show the most consistent spatial distribution of u > 0 across the study ecoregions." In the methods, it is stated that u has a range between 0 and 1 (line 294). Thus, to me the runoff datasets do not show a great consistency, since basically the whole range is still possible except 0 (based on the text). Another example is the description of figure 3c (line 355). I did not see 3 distinct clusters initially. However, maybe it is more easily visible if it is more visible how the different hydrologic models are clustered. Then, I have some suggestions for different figures to increase their clarity:

1) Fig. 3: continuing on the just made comment. I think it would be beneficial for your

analysis to indicate what type of model each hydrologic model is. Maybe a sub- or superscript can be added, e.g. 1 = LSM, 2 = CM and 3 = WBM.

2) Fig. 7: I have already mentioned this. One subplot is missing. Here a subplot concerning R can be added.

3) Fig. 8: The visibility of the 10th and 90th percentile lines is minimal. I would enhance this more.

4) Appendix 2.5 – 2.9: The dataset names overlap each other in multiple graphs. Also, these graphs are an addition to figure 6. However, these figures in the appendix are referred to more than figure 6 itself. To me, this seems odd.

5) Appendix 2.10: Here several bins are used. The first two are clear to me. I am not sure if I understood the last two completely. The bin with $p < 0.50$ does this mean 0 – 0.50 or 0.05 to 0.50? And the final bin with $p > 0.05$, should include most of the points of the previous bins. But these are not shown in the graph. That is why I though this bin is maybe $p < 0.05$. Have I understood this correctly?

6) General remark: There are quite a few figures in the whole study and most figures have quite a high information density. Because of this, I thought it was difficult to see where the research was heading. If possible, I would suggest to consider what the main results of this study are and focus on those results.

So to summarise, I would suggest to shorten the introduction; to structure the statistical methods more clearly; to adapt the case studies; and to thoroughly revise the analysis and use of figures.

Minor comments:

Besides these major comments, I have noticed some smaller issues. Most are quite easily changed.

1) The abstract is too long.

2) The datasets employed in this study were selected based on their public availability, which is I think a proper criterion to base the selection on. I did notice that the three types of datasets are not equally represented. Does this bias the results? Is it possible to reasonably compare the different types for 1 variable, e.g. for AET?

3) Throughout the main body of the manuscript, I noticed the abbreviation RZSM (root zone soil moisture) was used. Only in the introduction, the abbreviation SM (soil moisture) is used. I was initially confused because of this, since to me it seems to describe the same object. So I would recommend to be chose either one of the abbreviation for the whole manuscript, unless I have perhaps interpreted this wrongly.

4) Finally, the numbering of the sections and one table is incorrect. The methods section is numbered '3', which I think should be '2'. I also noticed that Table 2 has in its header number '1'. I would recommend to check the full manuscript for its numbering, both the sections and the figures/tables.

I look forward to reading the revised manuscript.

Kind regards, Janneke Remmers Wageningen University

---

## Referee Comment (RC2) · Nans Addor (Referee) · 19 Aug 2020

**Review of "Implications of Model Selection: A Comparison of Publicly Available, CONUS-Extent Hydrologic Component Estimates" by Saxe et al.**

This study compares the main components of the terrestrial water balance over the CONUS using a range of data products and model simulations. Not only do the authors provide a review of publicly available datasets, they also compare and cross-evaluate them. They use a range of approaches and indices (including innovative metrics like a measure of unalikeability) to explore the variability across these datasets and the resulting uncertainties in the water balance components. They show that significant gaps in the water balance can exist depending on which datasets are relied upon and they stress the importance of adopting a multi-model approach.

Overall, the analysis is thorough, pertinent and timely. The text is well written and the figures are carefully crafted. It is a challenge to summarise the results stemming from this variety of data sets, states/fluxes, time scales and hydroclimates, so some compromises have been made (e.g. seasonal variations are not explored and some ecoregions are quite large), but overall I support the decisions made by the authors. This study provides a rather comprehensive overview and comparison of publicly available and hydrologically relevant datasets, and hence constitutes a useful resource for future studies in the US and beyond. It fits within the scope of HESS. I congratulate the authors for this impressive data collection and analysis effort.

My comments below are mostly clarification requests and suggestions.

**Major comments**

CV I: The authors use the coefficient of variation (CV) to measure the variability among the different data products/models for each component of the water balance (Eq. 1). But while CV is provided for two 15-year periods, I understand that it is computed for each water year (L263), so it can potentially be used to quantify the inter-annual variability of each component (as suggested by the caption of Figure 2). Can the authors clarify if their intention with CV is to compute the inter-model spread and/or the inter-annual variability? If it is the former only, they might want to refer to the variability measured by CV as "uncertainty" to avoid confusion with (temporal) variability. Can they also clarify if/how CV values for individual water years were aggregated over 15 years?

CV II: One might argue that using CVs inflates the importance of (comparatively small) absolute differences between data products (e.g. in R, SWE and RZSM). For instance, the contribution of SWE [in mm] to the total water balance is small over the US (Fig 3d), and so is the SD among the SWE products (Fig 2a), yet the CV is quite large (Fig 2b). So, when it comes to reducing uncertainties and closing the water balance, should we first tackle components with large uncertainties (but potentially modest contribution to the water balance) or components with potentially smaller uncertainties but a larger contribution to the water balance? This is important when making recommendations in the abstract and conclusions. I see the value of the analysis as it currently stands, but to get a better sense of the uncertainty in each component with respect to the total water balance, it might be worth completing the analysis by computing a modified version of CV, using total input P as denominator

(similarly to Eq. 7) instead of x bar (Eq. 1). Showing these new results using a figure similar to Figure 5 could put the differences between datasets in the wider context of the water balance and possibly provide insights into the reasons behind the residuals shown in Figure 8. To some extent, this is already achieved by Figure A2.1, but the different units and ranges covered by the x-axes make component comparison difficult.

Sample size: What is the influence of the sample size on the CV estimates? There are fewer RZSM estimates than P, AET, R and SWE estimates, can this influence the conclusions? Can this be estimated by boostraping, e.g. by resampling with replacement using the same sample size for all the water balance components? This uncertainty could be shown in Figure 2.

Soil moisture: I agree that differences in RZSM can be related to the soil depth used by different models (L337), and in this respect, differences between models are less surprising than for the other state variables and fluxes. Still an important result, as hydrologists using a single model are likely to find it useful to know how their model compares to others. But why "rootzone" soil moisture (defined on L17 and used throughout the study) and not soil moisture? Is it because SM was computed only over the depth accessible by roots, and not over the whole soil thickness? Is the root depth/soil depth distinction made by all the models considered? Also, please comment on the soil depth actually covered by remote sensed products. Is it correct to say that the whole "rootzone" is covered, or is it less, which could bias soil moisture estimates and partially explain low correlations with hydrological model estimates (Figs. 7f and 7g)?

Water balance residuals: changes in groundwater storage are not quantified in this study, and instead, they are lumped into the water balance residual. The authors mention Gravity Recovery and Climate Experiment (GRACE) in the discussion but did not use GRACE data although it would have provided a first estimate of changes in terrestrial water storage and enabled them to go beyond soil moisture. Why was this key part of the water balance not characterised?

Eco-regions: please clarify why the Environmental Protection Agency Ecoregions were used instead of a classification more focussed on hydrology (e.g. Sawicz et al., 2011; Berghuijs et al, 2014; Knoben et al., 2018)?

Figure 7 provides an elegant overview of the correspondence between the remote-sensed data and the other datasets. Any ideas why MOD16-A2 and SSEBop correlate particularly poorly with the other datasets in the N. Amer. Desert and Medit. CA, respectively?

Uncertainties in all components of the terrestrial water balance exist and their existence is well documented (here and by others before). And many datasets are now publicly available (as highlighted here). Yet, studies that account for these uncertainties (e.g. using a range of data products) are still rare. What would you recommend to make such an approach more systematic? Can you suggest a way to select a subset of data sets for users who can't download and process all the datasets used here? Please also see my note at the end of this document.

**Minor comments**

L193: Currently there seems to be no section 2.

L608-610: What does the range provided for each variable correspond to? I assume there are retrieved from Figure 5. Please also clarify this in the abstract (L23-25)

Figure 2a: Is the SD for RZSMV really close to 0 (perfect agreement between models)?

Figure 7: maybe clarify whether each circle/square corresponds to one dataset and whether these datasets are sorted based on rho (and hence their order changes from one panel to the next).

Figure 8: "Histograms of eight ecoregion water budget relative residuals", maybe clarify that it is relative to the total precipitation (Eq 7).

Figure A2.5 to A2.9: I suggest
  - removing caption mentions to colours not used in the plot (e.g. green in A2.5)
  - clarifying whether the boxplots are made using significant tau values only
  - clarifying whether the dots represents significant tau values
  - re-evaluating whether box plots are well-adapted to show 1 or 2 values

**References**

Berghuijs, W. R., Sivapalan, M., Woods, R. A. and Savenije, H. H. G.: Patterns of similarity of seasonal water balances: A window into streamflow variability over a range of time scales, Water Resour. Res., 50, 5638–5661, doi:10.1002/2014WR015692, 2014.

Knoben, W. J. M., Woods, R. A. and Freer, J. E.: A Quantitative Hydrological Climate Classification Evaluated with Independent Streamflow Data, Water Resour. Res., 54, doi:10.1029/2018WR022913, 2018.

Sawicz, K., Wagener, T., Sivapalan, M., Troch, P. a. and Carrillo, G.: Catchment classification: Empirical analysis of hydrologic similarity based on catchment function in the eastern USA, Hydrol. Earth Syst. Sci., 15(9), 2895–2911, doi:10.5194/hess-15-2895-2011, 2011.

**Note to the authors**

I co-created the CAMELS dataset to facilitate access to hydrometeorological time series and landscape attributes of 671 CONUS catchments (see https://hess.copernicus.org/articles/19/209/2015/ and https://hess.copernicus.org/articles/21/5293/2017/). CAMELS is a community resource and we are looking for ways to expand it. Reviewing your manuscript, I thought that it would be great if time series for the CAMELS catchments could be extracted from the data products you downloaded and processed, and made publicly available. It would remove many barriers currently preventing water scientists and

practitioners from characterising uncertainties in the water balance. Feel free to contact me to discuss this further. Of course, your decision will not influence my review of the revised version of this study.

Best regards,

Nans Addor - University of Exeter, UK

---

## Author Response (AR1)

Subject:

Response to RC1

Text:

The authors are grateful for the invaluable input from the referee on our paper "Implications of Model Selection: A Comparison of Publicly Available, CONUS-Extent Hydrologic Component Estimates". The reviewers comments are insightful and identify to the authors areas where clarification would increase the effectiveness of the manuscript.

**Major Comments**

1. Regarding the readability of section 3.3:
   a. We agree that the use of subsections would increase the readability, as would explaining to the reader how each method contributes to the analysis and discussion of the manuscript.
2. Regarding the use of time periods for analysis:
   a. Time periods were used primarily to limit the effects of variable models counts within each water year. Datasets were typically available in either the (a) 1985-1999, (b) 2000-2014, or (c) 1985-2014 time periods, with counts of models available per year changing between water balance components. By dividing summary statistics into two time periods (Early vs. Late), we attempted to reduce biases that would be introduced into uncertainty values by having more models (i.e. greater uncertainty) or less models (i.e. less uncertainty). However, the other reviewer suggested using a bootstrapping methodology to calculate uncertainty that will further assist in reducing the biases. Using that method may allow us to remove the time periods.
3. Regarding the exclusion of two ecoregions from Figure 8:
   a. The two smallest ecoregions were excluded because, as the reviewer notes later in their comments, many of the figures in this manuscript are extremely informationally dense. In this case, we simply wanted to provide a more concise figure because it is used more to visually relate water balance uncertainty than it is to provide the reader with raw values.
4. Regarding more clearly highlighting the originality of the study:
   a. We agree that this can be better discussed within the paper. This will be amended during revisions.
5. Regarding line 294:
   a. Line 294 states that "Disagreement in the presence of significant trend and trend direction is quantified using the unalikeability coefficient (u) which measures how often categorical variables differ on a **$0 \le u \le 1$ scale**, with **0 and 1 being complete agreement and disagreement, respectively** (Kader and Perry, 2007). Thus, the value of 0 occurs if all datasets agree on trend direction. By stating on Line 464 that "Runoff datasets show the most consistent spatial distribution of u > 0 across the study ecoregions", we are explaining (in an unclear way) that runoff datasets are most commonly in disagreement across the CONUS. This will be edited for clarity in the revisions
6. Regarding the description of Figure 3c (line 355):

a. We agree that it would be useful to label each hydrologic model by its category (LSM vs. CM vs. WBM). This had actually been done earlier but discarded because the figures became too dense for the reader, but perhaps we can fit labels along the right-hand y-axis, or potentially shade the figure background by model type (e.g. white = LSM, light grey = CM, dark grey = WBM).

7. Regarding missing subplot in Figure 7:
   a. The empty space is left in place because we only have one remote sensing dataset for SWE and wanted each water balance component to have it's own row. We can't fill the empty space with a subplot of runoff (R) because there is not a satellite dataset that measures runoff, at least until SWOT is launched in 2022 and even then will not have retrospective estimates during our study period. This was not mentioned in the methods section discussing correlation statistics, so the revision will add it in for clarity.

8. Regarding Figure 8:
   a. The $10^{th}$ and $90^{th}$ percentile boxplot lines are not shown when they exceed the boundaries of the subplots. We were on the fence about whether to even include the boxplots as overlays on the histograms because they tend to "busy" the figure, so to speak, without providing much additional information. If parts of the boxplots have to be excluded in some subplots, perhaps it is best just to drop the boxplots entirely and leave the subplots as just histograms.

9. Regarding Appendix Figures 2.5-2.9:
   a. Yes, several of the dataset names overlap each other. This is something we will correct in the revisions. It was difficult to squeeze so much text into individual figures. We placed these figures into the appendix rather than the main manuscript body because they are quite large but difficult to shrink down considering all the included text. With Figure 6, our goal was to provide most readers with a simple visual representation of the general disagreement in model trends. Appendix Figures 2.5-2.9 were attached to provide more detailed results for readers that may be interested in specific models used in the study.

10. Regarding Appendix Figure 2.10:
    a. I believe you are misinterpreting the grouping labels in this figure. The first three groups, labeled as "Cor > 0.90", "Cor 0.50-0.90", and "Cor < 0.50", are grouping values of correlation measured with Spearman's rho, denoted in the text with the Greek letter $\rho$. Models are only assigned to these groups when their correlation is statistically significant, calculated using a binary p-value significance test. If the p-value of a significance test is less than our assumed alpha value ($\alpha$ = 0.05), then correlation is assumed to be significant. So the first group, "Cor > 0.90", identifies models with very strong significant correlation. The second group, "Cor 0.50-0.90", identifies models with moderate to good significant correlation. The third group, "Cor < 0.50", identifies models with poor to negative significant correlation, including anything with rho values of -1 to +0.50. The fourth group includes any model with statistically *insignificant* correlation but does not report the actual correlation value since it is deemed irrelevant by the significance test.

11. Regarding the general remark:

a. We agree that there are much more figures than are typically found in a journal paper, as well as being more informationally dense than usual. However, our goal with this study was to provide readers access to as much information as possible while still maintain a decent "readability" so that those readers interested in specific models can find the relevant information without having to delve into the actual datasets being released in tandem with this study. For example, we want the reader who is utilizing the NLDAS2-Noah land surface model in California to be able to compare their monthly and annual estimates to a range of other models without having to acquire, process, and interpret all the other models themselves.

b. Our hope is that these information-dense figures will allow the scientific community to more easily include uncertainty constraints in their results and analyses. We see this manuscript essentially as a review of the current state of knowledge within the various modeling communities measured in terms of uncertainty.

**Minor Comments**

1) Regarding abstract length: This will be shortened during revisions.
2) Regarding dataset types: Unfortunately, remote sensing datasets are much more limited than hydrologic model datasets. We try to discuss differences between dataset types in more general terms to soften potential biases resulting from different numbers of available datasets by water balance component. However, different remote sensing datasets estimating the same water balance component will likely use the same underlying observational measurements (e.g. MOD16-A2 and SSEBop both use MODIS sensor data). Because of this, we believe that comparing the magnitudes of just one or two remote sensing datasets against many hydrologically modeled datasets is effective in at least representing general differences.
3) Regarding the use of RZSM: This was also noted by the other reviewer. We will switch to using "SM" as an abbreviation during revisions.
4) Regarding numbering errors: This was also noted by the other reviewer. This will be corrected during revisions.

Subject:

Response to RC2

Text:

The authors are grateful for the valuable input from the referee on our paper "Implications of Model Selection: A Comparison of Publicly Available, CONUS-Extent Hydrologic Component Estimates". The major comments, especially, highlight suggestions that will enhance the relevance and clarity of our manuscript.

**Major comments**

**Regarding CV (I):** With the coefficient of variation (CV) metric, our primary goal is to quantify inter-model variability. Our secondary goal is to quantify temporal variability (inter-annual), though this is performed in only a rudimentary fashion by dividing datasets into "Early" and "Late" time periods. The use of CV is best exemplified in Figure 5. Confusion is likely arising from the early use of CV in Figure 2 where only a single value is presented for each component (e.g. precipitation, evapotranspiration, etc.). In the case of Figure 5, CV was calculated between all models for a single water year for each component for each ecoregion. Using Figure 5/component AET/ecoregion Medit. CA/period Early as an example, the data points used to generate the boxplot are the CV between all models for each water year from 1985-1999. We agree with the referee's suggestion to use the term "uncertainty" in place of CV to avoid confusion with temporal variability, this will be corrected in the revised manuscript. To answer the referee's question "Can they clarify if/how CV values for individual water years were aggregated over 15 years?": Data presented in Figure 5 show the spread of individual years. In the text, CV values are summarized over 15 year periods using the median CV measure.

**Regarding CV (II):** The referee makes a very valid point regarding CVs inflating the importance of relatively unimportant hydrologic components. We agree that that relating uncertainty relative to precipitation, rather than the underlying mean, would relate the importance of model uncertainty by hydrologic component. However, this is probably only be effective for flux components (precipitation, ET, runoff) and not storage components (SWE, soil moisture).

**Regarding sample size:** Agreed that uncertainty will increase as sample size increases. Bootstrapping using a constant sample size would provide an additional measure of uncertainty and would help to better inform readers.

**Regarding soil moisture:** The usage of the term "rootzone" was carried over from the first few models we included in this study (NLDAS and GLDAS land surface models). However, it is at best misleading and at worst incorrect to call the included soil moisture datasets "rootzone" products. We will amend the manuscript to use only "soil moisture". Remotely sensed products (ESA-CCI, SMOS-L4) cover different soil depth ranges, computed in terms of volumetric soil water content. ESA-CCI is an assimilation of data from various remote sensing instruments, while SMOS-L4 uses a double bucket water budget model to extrapolate surface soil moisture (0-5cm) to the "rootzone" domain (5-200cm). Again, though, the use here of the term "rootzone" is inaccurate and we will amend the manuscript accordingly.

**Regarding water balance residuals:** We used the term "imbalances" rather than "residuals" specifically to inform the reader that excess water, or lack thereof, in our water budget analyses is not a measure of accuracy. Rather, "imbalances" are used to show the reader that the presence and magnitude of gains or losses in a water budget equation are highly uncertain due to the uncertainty in terrestrial hydrology estimates, thereby showing the possible ranges in terrestrial water storage and groundwater storage. We excluded changes in groundwater storage from this study for a few reasons.

(1) Availability of modeled estimates: At the CONUS scale, there are few publicly available datasets to compare and use for estimates of uncertainty, relative to other components (e.g. precipitation, SWE). While we were offered results of a few models, those data were not available to the public, and thus not used in the context we discuss in the introduction. Furthermore, groundwater datasets covered a shorter time range than other datasets in this study and did not cover the conterminous U.S. This spatiotemporal heterogeneity would conflict with the homogeneity of the surface water flux and storage datasets used in this study, reducing the effectiveness of our uncertainty analysis.

(2) GRACE: Groundwater estimates are commonly derived from GRACE terrestrial water storage (TWS) data and provide an effective means of measuring changes in groundwater storage. However, to calculate changes in groundwater from GRACE TWS requires the subtraction of modeled data estimating soil moisture, snow, and surface water storage. We believe that this is outside the scope of this study as it would constitute a new analysis focusing on the implied uncertainty in groundwater derived from GRACE TWS. As a case in point, we are submitting a manuscript to another journal this week where we do just that: we provide a comprehensive analysis of uncertainty in GRACE-derived groundwater by quantifying how trend magnitude and direction are affected by model selection using ten surface water storage estimates (soil moisture and snow) and five GRACE TWS solutions (3 spherical harmonic and 2 mascon) over a seven year period.

**Regarding ecoregions:** The EPA ecoregions were used for classification for two reasons:

(1) Common use: the EPA ecoregions are commonly applied across a range of sub-disciplines within the hydrological, ecological, and geological scientific communities. They are also "understandable" in the sense that readers intuitively understand the differences between a region called "Eastern Temperate Forests" and "Northwestern Forested Mountains". With this paper, our target audience is not only the modeling community (e.g. land surface modelers, catchment modelers, remote sensing modelers) but also the users of model outputs who apply estimates to local and regional scale analyses who would otherwise be constrained in terms of technical abilities and computational resources from understanding and incorporating measures of uncertainty in their projects. We believe that using EPA ecoregions provides a useful, albeit highly simplified, classification system for intercomparing modeled hydrologic estimates.

(2) Dataset availability: The cited sources of classification schemes use somewhat limited sample sizes of 200-300 catchments (Sawicz et al., 2011; Berghuijs et al., 2014) that are disproportionately, or entirely, weighted towards the eastern CONUS. In all cases, the classification schemes are applied to cluster studied watersheds and not extrapolated over the CONUS, therefore not providing a continuous classification system that we could apply for our datasets.

Perhaps a useful future application of the datasets we collected for this study would be to extend the methods of Berghuijs et al., 2014 over the entirety of the CONUS to provide a multi-level classification system derived from hydrology rather than ecology.

**Regarding Figure 7 and correlation of datasets against MOD16 and SSEBop:**  It is likely that poor correlation between MOD16-A2 and other datasets in the North American Deserts ecoregion is caused by poor model performance of the MOD16-A2 dataset.  The MOD16-A2 dataset typically shows annual peak values occurring 1-3 months earlier than all other hydrologic and remote sensing models in the western area of the North American Deserts ecoregion.

In Mediterranean California, attributing poor correlation to either SSEBop or modeled datasets is more difficult.  On one hand, SSEBop typically has positive biases in regions with high levels of bare ground fraction, such as sparse shrubland areas common to the Mediterranean California ecoregion, which may affect the timing of seasonal trends.  On the other hand, bare ground fraction is even more common in the North American Desert ecoregion and yet correlation between SSEBop and other datasets is typically high.  A more likely cause of the poor correlation is that high levels of irrigation in the Central Valley of California may not be properly accounted for in the ET models, whereas the SSEBop model correctly identifies them due to it's use of remotely sensed MODIS data.  However, the MOD16-A2 dataset also uses MODIS data but shows better correlation with other datasets, further complicating the discussion. A deeper investigation into input parameters and calibration methods used in the hydrologic models would help to better understand these discrepancies.

**Regarding recommendations/suggestions:**  We strongly agree with the comment that studies accounting for uncertainty in modeled estimates are rare despite the prevalence of literature noting and quantifying these values. Previous Model Intercomparison Projects (MIPs), such as those discussed in the introduction (e.g. CMIP, WaterMIP, AgMIP, etc.), provide interesting formats to provide the scientific community with evaluation and validation measures. However, those MIPs are often either short-lived or limited in scope and don't provide measures of uncertainty that can be directly applied to individual studies. Our hope with this publication is that users of the study datasets can use our quantified uncertainty values within their own research to, at the very least, inform their subsequent readers of how uncertain their estimates of ET, P, etc. are and how that may impact results and conclusions.

A more systematic approach to providing measures of model uncertainty would be to compile a database in which a variety of modeled datasets (hydrologic, reanalysis, remote sensing models) are aggregated together in tandem with: (a) pre-computed intercomparison statistics, and/or (b) a GUI/app/R package/Python package to allow users to generate statistics based on their selected subsets of models. While collection of datasets is far easier today than 5-10 years ago, data are scattered across numerous databases that each require different access methods, permissions, output formats, coordinate reference systems, and spatiotemporal resolutions. To that end, we are tentatively planning a project to tackle this very issue, assuming that we can overcome the obstacles of data storage, GUI/app hosting costs, and dataset permissions.

In the context of this manuscript, we agree with the referee that the text should include examples of pre-existing sources of model subsets, such as the CAMELS database, to point readers in the direction of useful data. Additionally, all data created for this study (models aggregated by mean area weighting to 10 EPA Ecoregions at the monthly time step) will be made publicly available on a ScienceBase.gov page. Our revised manuscript will update the text to include these details.

**Minor Comments**

*"L193: Currently there seems to be no section 2."*

> Correct, this is a typo that will be amended in the revision.

*"L608-610: What does the range provided for each variable correspond to? I assume there are retrieved from Figure 5. Please also clarify this in the abstract (L23-25)"*

> The values provided for each variable are the ranges in coefficient of variation (CV) measurements for all ecoregions in either the western or eastern CONUS. For example, we state "Results show that flux and storage magnitudes disagree most greatly in the western CONUS, with CVs of precipitation (P) 11-22%...". In this case, we are saying that the CV of precipitation datasets for ecoregions in *western* half of the CONUS range from as low as 11% (found in North American Deserts) and as high as 22% (found in Temperate Sierras). This will be clarified both in the lines referenced here as well as in the abstract.

*"Figure 2a: Is the SD for RZSMV really close to 0 (perfect agreement between models)?"*

> This is the result of a difference in units between the datasets and an error in the vertical axis of the figure. Components P, AET, R, SWE, and RZSME are measured in equivalent water depth as millimeters. RZSMV is measured in volumetric water content fraction as m3/m3 (ranging from 0-1). So, the SD for RZSMV should be on a different vertical scale than the other components. This will be corrected in the manuscript revision.

*"Figure 7: maybe clarify whether each circle/square corresponds to one dataset and whether these datasets are sorted based on rho (and hence their order changes from one panel to the next)."*

> Agreed that this should be clarified in the figure caption. Indeed, each point, whether it be a circle or square, corresponds to a single dataset and are sorted based on rho. We included a more detailed figure in the appendix (A2.10) for interested readers.

*"Figure 8: "Histograms of eight ecoregion water budget relative residuals", maybe clarify that it is relative to the total precipitation (Eq 7)."*

> This will be corrected in the revision. While this is stated in the manuscript methods, clarification in the figure caption will be useful for readers and skimmers.

*"Figure A2.5 to A2.9: I suggest*

*- removing caption mentions to colours not used in the plot (e.g. green in A2.5)*

*- clarifying whether the boxplots are made using significant tau values only*

*- clarifying whether the dots represents significant tau values*

*- re-evaluating whether box plots are well-adapted to show 1 or 2 values"*

> We agree with all reviewer suggestions, and especially so on his last point. All suggestions will be addressed in the revisions.

**List of Changes**

- Terminology change:  Use term "uncertainty" instead of "variability" when discussing inter-model disagreement.
- Change evapotranspiration abbreviation from "AET" to "ET"
- Change abbreviation for soil moisture from "RZSM" to "SM" following suggestions by RC1 & RC2.
- Recalculated uncertainty measures (standard deviation and coefficient of variation) using a bootstrap methodology following the advice of RC2. Updated methods section 2.3.1 to describe and justify method.
- Updated CV and SD uncertainty values in the text following application of bootstrap method.
- Fixed typo in section numbering.
- Included additional subsections within Section 2.3 Statistics section to improve readability following suggestion of RC1.
- Clarified discussion in Section 3.2.1 to improve readability.
- Amended Section 4 Discussion to address existing model compilation datasets as suggested by RC2.
- Corrected component abbreviations in figures & tables.
- Updated Figure 2 to improve readability following suggestion of RC2.
- Updated Figure 3 to include notation identifying hydrologic model types (e.g. land surface models, catchment models, etc.) following suggestion of RC1.
- Amended Figure 7 caption to improve clarity following suggestion of RC2.
- Updated Figure 8 by removing overlain boxplots to improve clarity following suggestion of RC1.
- Amended Figure 8 caption to explain relative residuals, following suggestion of RC2.
- Remove Figures A2.5-A2.9 entirely and replaced with a single, simpler figure.  This followed the comments by RC1 (text overlay, unclear figure) and RC2 (boxplots not necessary when only 1-2 values present).
- Abstract length reduced slightly following suggestion by RC1.

[revised manuscript text omitted]

- **Field Code Changed**
- **Formatted:** Font: (Default) Calibri
- **Formatted:** Font: Times New Roman
- **Field Code Changed**
- **Formatted:** Font: (Default) Calibri
- **Formatted:** Font: (Default) Times New Roman
- **Formatted:** Font: (Default) Calibri
- **Field Code Changed**
- **Formatted:** Font: Times New Roman
- **Field Code Changed**
- **Formatted:** Font: (Default) Calibri
- **Formatted:** Font: (Default) Times New Roman
- **Field Code Changed**
- **Formatted:** Font: (Default) Calibri
- **Formatted:** Font: (Default) Times New Roman

| | | | | | |
|---|---|---|---|---|---|
| VegET[‡] | USGS | (Gabriel B. Senay, 2008)(Senay, 2008) | 1 km | 2000-2014 | AET, RZSM(e)E |

*Reanalysis*

| | | | | | |
|---|---|---|---|---|---|
| CanSISE | U. Toronto | (Mudryk & Derksen, 2017)(Mudryk and Derksen, 2017) | 1° | 1981-2010 | SWE |
| CMAP[†,‡] | CPC | (Xie & Arkin, 1997)(Xie and Arkin, 1997) | 2 ½° | 1979-present | P |
| DayMET[‡] | ORNL | (M. . Thornton et al., 2018)(Thornton et al., 2018) | 1 km | 1980-present | P, SWE |
| ERA5[‡] | ECMWF | (C3S, 2017) | ¼° | 1979-present | P |
| ERA5-Land | ECMWF | (C3S, 2019) | ¹/₁₀° | 2001-present | P |
| GPCC[‡] | GPCC | (Becker et al., 2013) | ½° | 1901-2013 | P |
| gridMET[‡] | U. of ID | (Abatzoglou, 2013) | ¹/₂₄° | 1979-present | P |
| Livneh et al. 2013[‡] | NOAA | (Livneh et al., 2013) | ¹/₁₆° | 1915-2011 | P |
| Maurer et al. 2002 | U. WA | (Maurer et al., 2002) | ⅛° | 1950-1999 | P |
| MERRA-Land[‡] | NASA | (Rienecker et al., 2011) | ½° | 1980-2016 | P |
| MERRA-2[‡] | NASA | (Gelaro et al., 2017) | ½° | 1980-present | P |
| NCEP-DOE[‡] | NCEP/DOE | (Kanamitsu et al., 2002) | 210 km | 1979-present | P |
| NLDAS2[‡] | NASA | (Y. Xia, NCEP/EMC, et al., 2009)(Xia, NCEP/EMC, et al., 2009) | ⅛° | 1979-present | P |
| PRISM[‡] | OSU | (PRISM Climate Group, 2004) | 4 km | 1895-present | P |
| Reitz et al. 2017[‡] | USGS | (Reitz et al., 2017) | 800 m | 2000-2013 | AET |
| UoD-v5[‡] | U. of DE | (Willmott & Matsuura, 2001)(Willmott and Matsuura, 2001) | ½° | 1950-1999 | P |
| WaterWatch[‡] | USGS | (Jian et al., 2008) | HU8 | 1901-present | R |

*Remote Sensing*

[revised manuscript text omitted]

**Comparison of Model Trends by Ecoregion**

**a) Precipitation**

τ ≥ 0.50
0 < τ < 0.50

Insignificant (p > 0.05)

-0.50 < τ < 0
τ ≤ -0.50

| Marine W. Coast Forest | Medit. CA | NW Forested Mtns | N. Amer. Deserts | So. Semiarid Highlands | Temp. Sierras | Great Plains | E. Temp. Forests | North. Forests | Trop. Forests |

**b) Evapotranspiration**

τ ≥ 0.50
0 < τ < 0.50

Insignificant (p > 0.05)

-0.50 < τ < 0
τ ≤ -0.50

| Marine W. Coast Forest | Medit. CA | NW Forested Mtns | N. Amer. Deserts | So. Semiarid Highlands | Temp. Sierras | Great Plains | E. Temp. Forests | North. Forests | Trop. Wet Forests |

**c) Runoff**

τ ≥ 0.50
0 < τ < 0.50

Insignificant (p > 0.05)

-0.50 < τ < 0
τ ≤ -0.50

| Marine W. Coast Forest | Medit. CA | NW Forested Mtns | N. Amer. Deserts | So. Semiarid Highlands | Temp. Sierras | Great Plains | E. Temp. Forests | North. Forests | Trop. Wet Forests |

**d) Snow Water Equivalent**

τ ≥ 0.50
0 < τ < 0.50

Insignificant (p > 0.05)

-0.50 < τ < 0
τ ≤ -0.50

| Marine W. Coast Forest | Medit. CA | NW Forested Mtns | N. Amer. Deserts | So. Semiarid Highlands | Temp. Sierras | Great Plains | E. Temp. Forests | North. Forests | Trop. Wet Forests |

**e) Soil Moisture (equivalent water depth)**

τ ≥ 0.50
0 < τ < 0.50

Insignificant (p > 0.05)

-0.50 < τ < 0
τ ≤ -0.50

| Marine W. Coast Forest | Medit. CA | NW Forested Mtns | N. Amer. Deserts | So. Semiarid Highlands | Temp. Sierras | Great Plains | E. Temp. Forests | North. Forests | Trop. Wet Forests |

**f) Soil Moisture (volumetric content)**

τ ≥ 0.50
0 < τ < 0.50

Insignificant (p > 0.05)

-0.50 < τ < 0
τ ≤ -0.50

[revised manuscript text omitted]

---

## Author Response (AR2)

**Minor revisions, Author response to RC #1.**

1. Regarding section 2.3.4, section 3.2.5 and also Fig. 8, I made a remark concerning the use of imbalances during the last review. This was not addressed. I said the following:

"Concerning the imbalances, the terminology of 'imbalances' is proper, as opposed to using 'residuals'. The implicit assumption here is that the residuals, i.e. model uncertainty, are considered constant for every case study. Clark et al. (2008) show that model uncertainty differs between climates. Especially, more arid climate have higher residuals. Thus, the analysis based on the relative imbalance can only be semi-reasonably compared within 1 ecoregion. I do think imbalances is an interesting concept to use."

Clark, M. P., Slater, A. G., Rupp, D. E., Woods, R. A., Vrugt, J. A., Gupta, H. V., ... & Hay, L. E. (2008). Framework for Understanding Structural Errors (FUSE): A modular framework to diagnose differences between hydrological models. Water Resources Research, 44(12).

In the current version of the paper, this point is still valid, because this assumption is not mentioned in section 2.3.4. However, the authors still compare the different ecoregions in section 3.2.5. The results showed in the subplots of Fig. 8 are compared in section 3.2.5.

Response: I'm not understanding this comment, specifically the line "The implicit assumption here is that residuals, i.e. model uncertainty, are considered constant for every case study". I don't think this is a correct interpretation. We are not assuming that model uncertainty is constant between climates, in fact we are comparing *how* spatially varying uncertainty affects interpretations and conclusions drawn from a simple water balance analysis.

1. In my opinion, the abstract is still too long. I noticed you have shortened it already, well done. At the moment the results are listed in detail. Perhaps, it is possible to summarise this more. It would be great if you can write the abstract in one paragraph.

Response: We agree. The abstract has been trimmed by about 40%, now fitting into a single paragraph.

2. The introduction is 4.5 pages. This is quite long. When I first read the introduction, my attention waned from line 70 onwards and I skipped ahead till the end of the introduction. For me, the introduction became interesting again from line 168. The section that I skipped contained a lot of background information regarding what is already done in literature. I understand that for a CSA study, it is important to also show what the current state of affairs is in literature. So, I agree that this should be placed in the article. I just wonder if this should be placed in the introduction. I would recommend to add another section after the introduction that details this information: "Current state of affairs in literature" or some other title. Since this new section would still be about 3 pages long, I would also recommend to divide this new section into several subsections to increase readability.

Response: Rather than insert an entirely new section, we have significantly reduced the overly detailed comparison/validation literature review in the introduction to a more readable and succinct form.

3. For Fig. 7, thank you for the explanation. I understand now why there is no plot for runoff (no satellite data available). I would still recommend moving SWE to the bottom and removing the empty plot. I think leaving this in would only raise questions. If there is an empty blank space in the bottom right, it will not disrupt the overall look of the figure as much.

Response: The figure has been corrected.

4. Concerning Fig. 8, I agree with the reasons (the graphs are already information dense) the authors have given for leaving out 2 ecoregions. What I still do not understand though is the threshold for excluding these two, since the difference between the "Southern semi-arid highlands" ecoregion, which is excluded, and the "Temperate sierras" ecoregion, which is included, is only 0.29% of the whole CONUS land surface. Can this be clarified further?

Response: We have added in the two excluded regions to Figure 8, as well as corrected typos in the plot and x-axis titles where "Residual" was used instead of "Imbalance". Table 4 was updated to include summary statistics for the two additional regions.

5. Finally, in Fig. 2 panel b) Coefficient of variation has an 'a' in front of it in the graph. This should be changed to a 'b'.

Response: This has been corrected.

Extraneous figures in the Supplement were removed as well.